EMBO
Molecular Medicine

# Dietary magnesium supplementation improves lifespan in a mouse model of progeria

Ricardo Villa-Bellosta[*] (ID)

## Abstract

Aging is associated with redox imbalance according to the redox theory of aging. Consistently, a mouse model of premature aging ($Lmna^{G609G/+}$) showed an increased level of mitochondrial reactive oxygen species (ROS) and a reduced basal antioxidant capacity, including loss of the NADPH-coupled glutathione redox system. $Lmna^{G609G/+}$ mice also exhibited reduced mitochondrial ATP synthesis secondary to ROS-induced mitochondrial dysfunction. Treatment of $Lmna^{G609G/+}$ vascular smooth muscle cells with magnesium-enriched medium improved the intracellular ATP level, enhanced the antioxidant capacity, and thereby reduced mitochondrial ROS production. Moreover, treatment of $Lmna^{G609G/+}$ mice with dietary magnesium improved the proton pumps (complexes I, III, and IV), stimulated extramitochondrial NADH oxidation and enhanced the coupled mitochondrial membrane potential, and thereby increased $H^+$-coupled mitochondrial NADPH and ATP synthesis, which is necessary for cellular energy supply and survival. Consistently, magnesium treatment reduced calcification of vascular smooth muscle cells *in vitro* and *in vivo*, and improved the longevity of mice. This antioxidant property of magnesium may be beneficial in children with HGPS.

**Keywords** aging; HGPS; magnesium; progeria; vascular calcification
**Subject Categories** Metabolism; Pharmacology & Drug Discovery

## Introduction

Hutchinson–Gilford progeria syndrome (HGPS) is an extremely rare, sporadic genetic disorder that is characterized by premature aging and accelerated cardiovascular disease progression, including that of vascular calcification(Nair *et al*, 2004; Salamat *et al*, 2010; Hanumanthappa *et al*, 2011). Most HGPS patients carry a *de novo* non-inherited autosomal dominant heterozygous mutation of the *LMNA* gene (p.G608G in humans; p.G609G in mice) (De Sandre-Giovannoli *et al*, 2003; Eriksson *et al*, 2003). This mutation activates a cryptic splice donor site, which causes synthesis of a lamin A mutant that disrupts nuclear membrane architecture and induces multiple cellular defects, including abnormal gene transcription, signal transduction, and DNA damage. HGPS patients die at a mean age of 13–14 years (a mean of ~38 weeks old in $Lmna^{G609G/+}$ mice), typically because of a cardiovascular event (Merideth *et al*, 2008).

Experimental and observational studies have shown that high magnesium intake has beneficial effects on cardiovascular risk factors, mediated by improvements in insulin-glucose metabolism, endothelium-dependent vasodilation, and the lipid profile, a reduction in vascular calcification, and the induction of anti-hypertensive and anti-inflammatory effects (DiNicolantonio *et al*, 2018; Rosique-Esteban *et al*, 2018). For example, vascular calcification in uremic rats is prevented by magnesium supplementation (Diaz-Tocados *et al*, 2017). However, magnesium also plays diverse roles in the pathogenesis of cardiovascular diseases at the biochemical and cellular levels (DiNicolantonio *et al*, 2018; Rosique-Esteban *et al*, 2018).

Magnesium is an essential mineral that serves as a cofactor in more than 300 enzymatic reactions, including those involved in energy metabolism and protein/nucleic acid synthesis. Magnesium is essential for mitochondrial function and particularly for ATP production, and magnesium deficiency is found in cardiovascular disease, type 2 diabetes mellitus, hypertension, heart failure, and ventricular arrhythmia patients (DiNicolantonio *et al*, 2018; Rosique-Esteban *et al*, 2018). In addition, magnesium supplementation improves mitochondrial and cardiac diastolic function in diabetic patients (Liu *et al*, 2019).

Vascular calcification has been identified in a mouse model of Hutchinson–Gilford progeria syndrome (Villa-Bellosta *et al*, 2013). The excessive accumulation of calcium in the vessels of HGPS mice (Osorio *et al*, 2011) is associated with defective extracellular pyrophosphate metabolism, due to a reduction in ATP synthesis secondary to mitochondrial dysfunction (Villa-Bellosta *et al*, 2013).

In the present study, we aimed to determine whether magnesium supplementation ameliorates vascular calcification and improves longevity in $Lmna^{G609G/+}$ mice.

## Results

### Magnesium improves $Lmna^{G609G/+}$ vascular smooth muscle cell (VSMC) viability

Several studies have shown that the accumulation of DNA damage in cells activates DNA damage and replication checkpoints, which

Fundación Instituto de Investigación Sanitaria, Fundación Jiménez Díaz, Universidad Autónoma de Madrid, Madrid, Spain
*Corresponding author. Tel: +34 91 550 48 97; E-mail: metabol@hotmail.com

attenuate cell-cycle progression and arrest replication (Liu *et al*, 2005, 2006; Varela *et al*, 2005; Richards *et al*, 2011; Sieprath *et al*, 2015). We first performed a comparative analysis of the proliferative ability of primary vascular smooth muscle cells from *Lmna*$^{G609G/+}$ mice and their wild-type littermates. Notably, microscopy images showed an apparent similar cellular morphology in both genotypes during its growth (Fig EV1A). However, *Lmna*$^{G609G/+}$ VSMCs exhibited much lower proliferation than control cells (Fig EV1B). The rate of division per day was significantly lower (by 36%) than that of wild-type control cells (0.36 ± 0.07 versus 0.23 ± 0.06 divisions per day; Fig EV1C; Appendix Table S1).

To determine the status of DNA replication, the replicative incorporation of 5-bromodeoxyuridine (BrdU) was assessed (Fig EV1D; Appendix Table S1). DNA synthesis in *Lmna*$^{G609G/+}$ VSMCs occurred at a 44% slower rate than in wild-type cells. Notably, *Lmna*$^{G609G/+}$ VSMCs incubated in medium containing a high magnesium concentration showed a significantly higher replication rate, both with respect to the number of divisions per day (0.30 ± 0.05), and the replicative incorporation of BrdU (75% of wild type).

Cellular activity, measured as cellular mitochondrial dehydrogenase activity, was significantly lower (by 35%) in *Lmna*$^{G609G/+}$ VSMCs than in control cells (Fig EV1E; Appendix Table S1). Moreover, *Lmna*$^{G609G/+}$ VSMCs had significantly lower (40%) intracellular ATP concentrations versus control cells (Fig EV1F; Appendix Table S1). In addition, senescence-associated β-galactosidase (β-gal) activity was significantly higher (3-fold) in *Lmna*$^{G609G/+}$ VSMCs than in wild-type cells (Fig EV1G; Appendix Table S1).

Notably, *Lmna*$^{G609G/+}$ VSMCs treated with magnesium-enriched medium showed significantly higher intracellular ATP (24%) and cellular activity (21%) than untreated *Lmna*$^{G609G/+}$ VSMCs. In contrast, *Lmna*$^{G609G/+}$ VSMCs treated with magnesium-enriched medium showed significantly lower β-gal activity (33%) than untreated *Lmna*$^{G609G/+}$ VSMCs.

### Magnesium improves mitochondrial ATP synthesis in *Lmna*$^{G609G/+}$ VSMCs

Previous studies have demonstrated mitochondrial dysfunction in progeria (Rivera-Torres *et al*, 2013; Villa-Bellosta *et al*, 2013; Aliper *et al*, 2015). Both oxygen consumption ratio (OCR) and ATP synthesis were significantly lower (by 41% and 39%, respectively) in *Lmna*$^{G609G/+}$ VSMCs than in wild-type cells (Fig 1A and B; Appendix Table S2). Moreover, *Lmna*$^{G609G/+}$ VSMCs had significantly lower mitochondrial membrane potential (ΔΨ$_m$; 37%), assessed using the red-to-green ratio of JC-10 fluorescence, than wild-type cells (Fig 1C; Appendix Table S2). Notably, *Lmna*$^{G609G/+}$ VSMCs showed significant higher ΔΨ$_m$ (32%), OCR (37%), and mitochondrial ATP synthesis (31%) when incubated in a magnesium-enriched medium.

### Magnesium ameliorates mitochondrial oxidative stress in *Lmna*$^{G609G/+}$ VSMCs

Mitochondrial reactive oxygen species (ROS)-mediated cell damage has been implicated in progeria (Richards *et al*, 2011; Sieprath *et al*, 2015; Kadoguchi *et al*, 2020). To evaluate the antioxidant properties of magnesium, ROS concentration was measured using the cell permeant reagent 2′,7′-dichlorofluorescin diacetate (DCFDA), a fluorogenic dye that can be used to quantify hydroxyl, peroxyl, and other ROS activities within the cell. *Lmna*$^{G609G/+}$ VSMCs showed significantly higher (3-fold) ROS content than wild-type cells (Fig EV2A; Appendix Table S3). In addition, the concentrations of two specific ROSs were also assessed. Mitochondrial superoxide (O$_2^-$) and hydrogen peroxide (H$_2$O$_2$) were present in significantly higher (1.6-fold and 2.3-fold, respectively) concentrations in *Lmna*$^{G609G/+}$ VSMCs than in wild-type cells (Fig EV2B; Appendix Table S3). Notably, this overproduction of ROS was significantly reduced (by 69% for ROS, by 43% for H$_2$O$_2$, and by 29% for O$_2^-$) in *Lmna*$^{G609G/+}$ VSMCs incubated in magnesium-enriched medium.

The rate of ROS generation and the cellular defenses against ROS toxicity (which include enzymes, small molecules, and proteins) contribute to the overall level of oxidative stress. The total antioxidant capacity (TAC) can be considered a cumulative index of antioxidant status. To evaluate the overall cellular capacity to counteract ROS, TAC was assessed using a Cu$^{2+}$ reduction assay. *Lmna*$^{G609G/+}$ VSMCs showed significantly lower TAC (38%) than wild-type VSMCs (Fig EV2C; Appendix Table S3). This reduction was significantly ameliorated (by 27%) in *Lmna*$^{G609G/+}$ VSMCs incubated in magnesium-enriched medium.

Reduced glutathione (GSH) is the major detoxifying redox buffer in cells and participates in the defense against ROS and the repair of mitochondrial oxidative damage, by being both a potent antioxidant itself and a substrate for antioxidant enzymes, including the glutathione reductase redox systems. Notably, total glutathione,

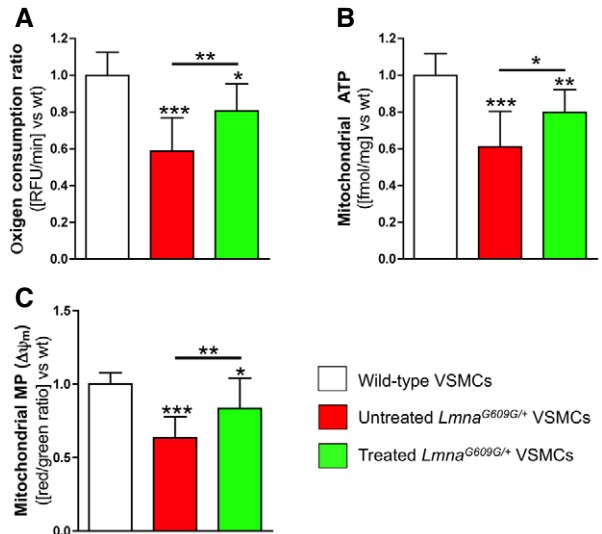

**Figure 1. Magnesium improves ATP synthesis in *Lmna*$^{G609G/+}$ VSMCs.**

A–C  (A) Oxygen consumption ratio, (B) mitochondrial ATP synthesis, and (C) mitochondrial membrane potential (MP), in the indicated VSMC types. Results are presented as the mean ± SD of three independent experiments (four wells *per* experiment). One-way ANOVA and Tukey's multiple comparisons *post hoc* test were used for statistical analysis. *$P$ < 0.05; **$P$ < 0.01; ***$P$ < 0.001.

Source data are available online for this figure.

which includes GSH and oxidized glutathione (GSSG), and glutathione reductase (GR) activity were significant lower in $Lmna^{G609G/+}$ VSMCs than in wild-type cells (Fig EV2D and E; Appendix Table S3). In addition, the ratio of reduced glutathione to oxidized glutathione (GSH:GSSG) was measured to assess the oxidative profile of the cells. $Lmna^{G609G/+}$ VSMCs showed a significantly lower (51%) GSH:GSSG ratio than wild-type VSMCs (Fig EV2D). Notably, this reduction was significantly ameliorated (by 51%) in treated $Lmna^{G609G/+}$ VSMCs, although the GR activity and total glutathione concentration were similar in treated and untreated $Lmna^{G609G/+}$ VSMCs.

GR uses reduced nicotinamide adenine dinucleotide phosphate (NADPH) to maintain the GSH redox state. Notably, although both types of VSMCs contained similar amounts of total nicotinamide adenine dinucleotide phosphate (NADPH and its oxidized form, $NADP^+$), $Lmna^{G609G/+}$ VSMCs had a significantly lower (48%) $NADPH:NADP^+$ ratio than wild-type cells (Fig EV2F and G; Appendix Table S3). However, the $NADPH:NADP^+$ ratio was significantly improved (by 45%) by magnesium treatment of $Lmna^{G609G/+}$ VSMCs.

## Magnesium ameliorates acidification-induced mitochondrial calcium overload

An increase in glycolysis that compensates for the loss of mitochondrial ATP synthesis has previously been shown in patient cells (Rivera-Torres *et al*, 2013). $Lmna^{G609G/+}$ VSMCs showed higher cytosolic ATP synthesis (1.8-fold, Fig 2A; Appendix Table S4), lactate production (1.9-fold, Fig 2B; Appendix Table S4), and extracellular acidification (2.1-fold, Fig 2C; Appendix Table S4) than wild-type cells. However, $Lmna^{G609G/+}$ VSMCs incubated in a magnesium-enriched medium showed significantly lower intracellular lactate concentration (19%) and extracellular acidification (14%). In contrast, cytosolic ATP synthesis was 21% higher in treated $Lmna^{G609G/+}$ VSMCs than in untreated cells. This result is consistent with the notion that magnesium increases the activities of the ATP-coupled glycolytic enzymes hexokinase, phosphofructokinase, phosphoglycerate kinase, and pyruvate kinase (Pilchova *et al*, 2017).

Intracellular acidification can lead to cytosolic and mitochondrial calcium overload, which depolarizes $\Delta\Psi_m$ to limit ATP production and stimulates mitochondrial ROS generation and permeability transition (Brookes *et al*, 2004; Görlach *et al*, 2015; Santulli *et al*, 2015). $Lmna^{G609G/+}$ VSMCs incubated with $^{45}Ca^{2+}$ as a radiotracer showed significantly higher (55%) mitochondrial calcium than wild-type cells, which was significantly reduced (by 21%) in treated $Lmna^{G609G/+}$ VSMCs (Fig 2D). In addition, $Lmna^{G609G/+}$ VSMCs showed significantly lower (35%) mitochondrial magnesium than wild-type cells, which was significantly increased (by 37%) in treated $Lmna^{G609G/+}$ VSMCs (Fig 2E and F).

## Magnesium prevents phosphate-induced $Lmna^{G609G/+}$ VSMC calcification

Previous studies show that calcification can occur without cellular activity, both in cultured devitalized aortas (Villa-Bellosta, 2018) and in fixed smooth muscle cells (Villa-Bellosta & Sorribas, 2009; Villa-Bellosta *et al*, 2011). To determine the effect of magnesium

on vascular calcification, treated and untreated $Lmna^{G609G/+}$ VSMCs were incubated in 2 mM phosphate-calcifying medium. Phosphate-induced calcification was then assessed in both living and fixed cells (Fig 3A–G; Appendix Table S5). Untreated $Lmna^{G609G/+}$ VSMCs showed 11-fold higher (in live cells) and 17-fold higher (in fixed cells) calcium deposition after 7 days of incubation in phosphate-calcifying medium. However, treated living $Lmna^{G609G/+}$ VSMCs showed significantly lower calcium accumulation (7.3-fold), although the calcium content in fixed cells was similar in treated and untreated $Lmna^{G609G/+}$ VSMCs (17-fold). The addition of pyrophosphate or phosphonoformic acid (two known inhibitors of calcium phosphate crystal deposition) (Villa-Bellosta & Sorribas, 2009) to the phosphate-calcifying medium completely prevented calcium accumulation in both fixed/living and treated/untreated $Lmna^{G609G/+}$ VSMCs. Notably, magnesium supplementation of the phosphate-calcifying medium significantly reduced (by 38% in untreated and 51% in treated cells) calcium deposition in living cells. By contrast, magnesium supplementation did not reduce calcium deposition in either treated or untreated fixed VSMCs. Taken together, these results suggest that magnesium prevents calcium phosphate deposition by a cellular activity-dependent mechanism, and not by direct binding to calcium phosphate crystals, preventing their formation and growth. Finally, the capacity to inhibit calcification ($\Delta Ca^{2+}$) was calculated as the difference in calcium deposition in living versus fixed cells ($Ca^{2+}$ in fixed cells − $Ca^{2+}$ in living cells). The $\Delta Ca^{2+}$ in treated cells was significantly higher than that in untreated cells (Fig 3G). Importantly, magnesium supplementation of the phosphate-calcifying medium caused significant increases in $\Delta Ca^{2+}$ in both treated and untreated cells.

## Magnesium prevents vascular calcification in HGPS mice

Clinically, plasma magnesium is usually measured despite the fact that less than 1% of magnesium exists extracellularly. Hence, plasma magnesium levels do not always accurately reflect total body magnesium stores. In fact, plasma magnesium levels can be normal despite depletion of the total body magnesium content. Notably, plasma magnesium levels were in the normal range in both wild-type and $Lmna^{G609G/+}$ mice, although they were significantly lower in 21- and 34-week-old $Lmna^{G609G/+}$ mice than in wild-type littermates (Table EV1).

To assess the effect of supplemental magnesium on $Lmna^{G609G/+}$ mice, their drinking water was supplemented with $MgCl_2$. Thereafter, the consumption of food and water was measured in the mice between 8 and 34 weeks of age. The median food and water consumption of untreated and treated $Lmna^{G609G/+}$ mice was similar ($3.46 \pm 0.77$ versus $3.53 \pm 0.72$ g/day/mouse and $3.96 \pm 0.62$ versus $4.01 \pm 0.73$ ml/day/mouse, respectively). Therefore, the total magnesium intake by treated $Lmna^{G609G/+}$ mice was significantly higher (4.6-fold) than that by untreated $Lmna^{G609G/+}$ mice ($976.2 \pm 261.7$ versus $213.9 \pm 45.0$ mg/day/kg; Fig 4A, see Materials and Methods section). Notably, the plasma magnesium concentration was significantly higher in treated $Lmna^{G609G/+}$ mice than in untreated $Lmna^{G609G/+}$ mice ($1.02 \pm 0.06$ versus $0.96 \pm 0.05$ mM; Fig 4B; Appendix Table S6). Finally, the total calcium content of aortas obtained from treated $Lmna^{G609G/+}$ mice was significantly lower than that of aortas obtained from untreated $Lmna^{G609G/+}$

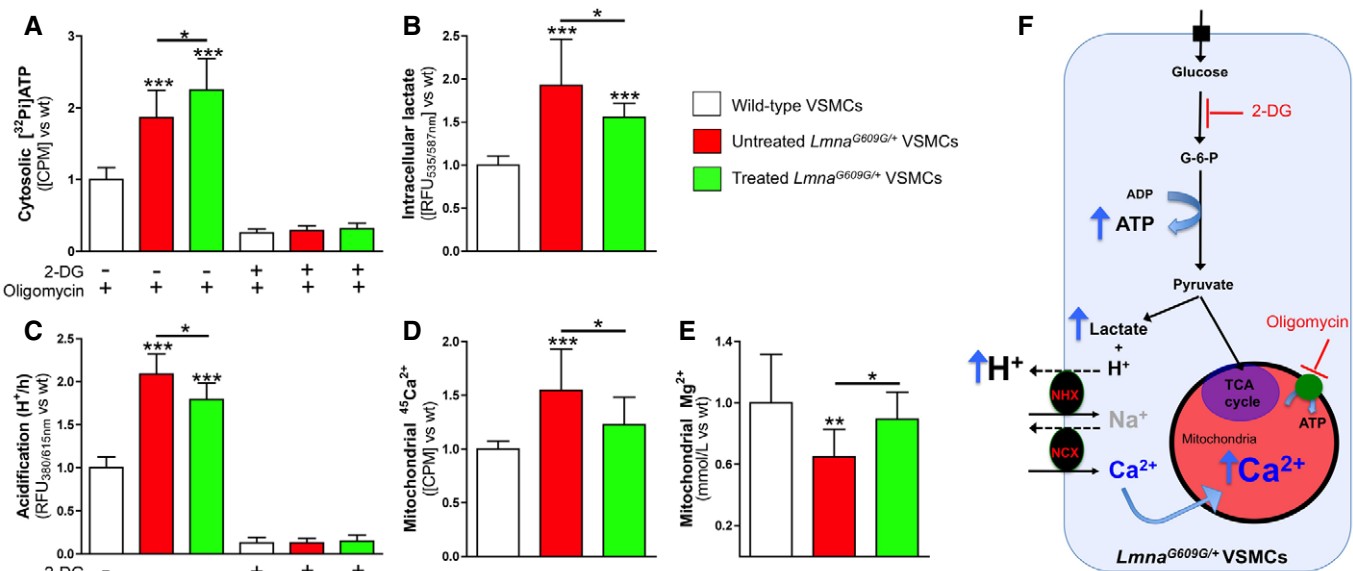

**Figure 2. Magnesium reduces acidification-induced mitochondrial calcium overload.**

A   Cytosolic ATP synthesis in the indicated VSMC types, measured by incorporation of phosphate-32 ($^{32}$Pi) into ADP. [$^{32}$Pi]-ATP was separated from $^{32}$Pi using the molybdate method, as explained in the Materials and Methods section.

B, C   (B) Intracellular lactate concentration and (C) external acidification in the indicated VSMC types.

D   Calcium accumulation in mitochondria after 24 h of incubation in MEM containing 10 μCi/ml calcium-45 ($^{45}$Ca$^{2+}$) as a radiotracer.

E   Magnesium concentration in isolated mitochondria.

F   The boxed scheme describes the mitochondrial calcium overload hypothesis. Lactic acidosis forces the Na$^+$/H$^+$ exchanger (NHX) to import Na$^+$, resulting in cytosolic Na$^+$ overload. Subsequently, the Na$^+$/Ca$^{2+}$ exchanger (NCX) is forced into reverse mode to dispose of excess Na$^+$, resulting in cytosolic calcium overload. This Ca$^{2+}$ is then taken up by mitochondria, resulting in mitochondrial calcium overload. 2-DG (2-deoxyglucose; 50 mM) blocks glycolysis through competitive hexokinase inhibition, whereas oligomycin (10 μM) inhibits mitochondrial ATP synthase. G-6-P: glucose-6-phosphate.

Data information: Results are presented as the mean ± SD of three independent experiments (four wells per experiment). One-way ANOVA and Tukey's multiple comparisons *post hoc* test were used for statistical analysis. *$P < 0.05$; **$P < 0.01$; ***$P < 0.001$.

Source data are available online for this figure.

mice (401.5 ± 77.7 versus 741.9 ± 101.6 μg/g aorta; Fig 4C; Appendix Table S6).

## Magnesium improves the longevity of HGPS mice

The body mass of 34-wk-old treated *Lmna*[G609G/+] mice was significantly higher (10%) than that of untreated *Lmna*[G609G/+] mice (26.5 ± 1.1 versus 24.0 ± 2.4 g; Fig 4D). Moreover, the median survival time of treated *Lmna*[G609G/+] mice was extended from 38.2 weeks to 42.9 weeks (Fig 4E and F).

## Magnesium improves the antioxidant status of HGPS mice

Liver homogenates from *Lmna*[G609G/+] mice had 42% lower TAC (Fig EV3A; Appendix Table S7), 38% lower total glutathione (Fig EV3B; Appendix Table S7), 48% lower GSH:GSSG ratio (Fig EV3C; Appendix Table S7), 55% lower NADPH:NAD$^+$ ratio (Fig EV3D; Appendix Table S7), and 43% lower GR activity (Fig EV3E; Appendix Table S7) than wild-type mice, implying the presence of an impairment in the NADPH-coupled GR redox system. Notably, treated *Lmna*[G609G/+] mice showed significant improvements in TAC (26%), GSH:GSSG ratio (52%), and NADPH:NAD ratio (45%) compared with untreated *Lmna*[G609G/+] mice. However, total glutathione and GR activity were not significantly better in treated *Lmna*[G609G/+] mice versus untreated *Lmna*[G609G/+] mice.

## Magnesium improves ATP synthesis in HGPS mice

Liver homogenates from untreated *Lmna*[G609G/+] mice showed significantly lower (55%) intracellular ATP, which was 65% higher in treated mice (Fig 5A; Appendix Table S8). Moreover, isolated mitochondria showed 89% higher calcium content in untreated *Lmna*[G609G/+] mice relative to wild-type mice, but this was 34% lower in treated mice (Fig 5B; Appendix Table S8). In contrast, isolated mitochondria showed 33% lower magnesium content in untreated *Lmna*[G609G/+] mice relative to wild-type mice, but this was 35% higher in treated mice (Fig 5C; Appendix Table S8). Moreover, the activities of complexes I, III, IV, and V were significantly lower in untreated *Lmna*[G609G/+] than wild-type mice, but these defects were significantly ameliorated in treated *Lmna*[G609G/+] mice (Fig 5D and E; Appendix Table S8), Notably, the subunits of these mitochondrial complexes are encoded by mitochondrial DNA.

## Magnesium improves mitochondrial ATP synthesis

ATP synthesis in isolated mitochondria was significantly lower in untreated *Lmna*[G609G/+] mice than in wild-type mice, in media containing either 0.1 mM magnesium (109.9 ± 29.6 versus 244.8 ± 78.8 nmol/min/mg protein, respectively) or 1 mM magnesium (226 ± 75.1 versus 503.7 ± 104.4 nmol/min/mg protein,

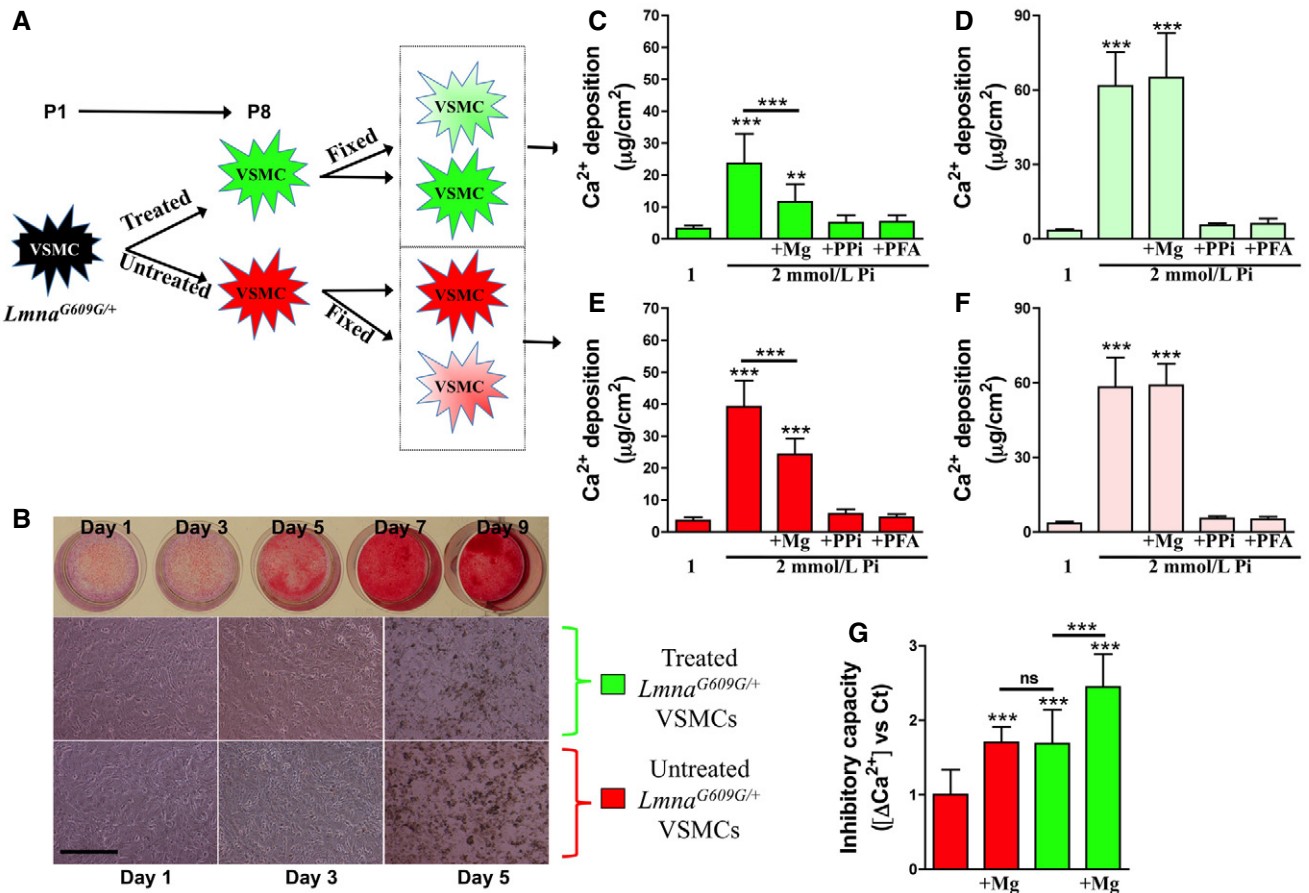

**Figure 3. Magnesium improves *Lmna*^G609G/+^ vascular smooth muscle cell calcification.**

A   Scheme showing the principle of the measurement. *Lmna*^G609G/+^ VSMCs were incubated in MEM (containing 0.8 mM magnesium; untreated) or in magnesium-enriched MEM (containing 1.8 mM magnesium; treated) from passage 1 to passage 8. Then, cells were incubated overnight in MEM containing 0.1% FBS and some cells were fixed, as described in the Materials and Methods section. Then, cells were incubated in MEM (containing 0.1% FBS) with 1 or 2 mM phosphate, 1.8 mM magnesium (+Mg), 100 µM pyrophosphate (+PPi), or 500 µM phosphonoformic acid (+PFA). After 7 days of incubation, during which the media were replaced daily, the calcium content was measured as described in the Materials and Methods section.

B   Representative time-course of 2 mM phosphate on calcification of *Lmna*^G609G/+^ VSMCs (up). Calcification was visualized with Alizarin red. Representative microscopic images (10x; scale bar: 100 µm) showing calcification of treated and untreated *Lmna*^G609G/+^ VSMCs (down).

C–F   Measures of calcium in treated living *Lmna*^G609G/+^ VSMCs (C), treated fixed *Lmna*^G609G/+^ VSMCs (D), untreated living *Lmna*^G609G/+^ VSMCs (E), and untreated fixed *Lmna*^G609G/+^ VSMCs (F).

G   The calcification inhibitory capacity was calculated as the difference in calcium deposition between living and fixed cells ($\Delta Ca^{2+}$).

Data information: Results are presented as the mean ± SD of three independent experiments (four wells per condition). One-way ANOVA and Tukey's multiple comparisons *post hoc* test were used for statistical analysis. **$P < 0.01$; ***$P < 0.001$.

Source data are available online for this figure.

---

respectively; Fig EV4A; Appendix Table S9). In both media, ATP synthesis was significantly higher (57% and 54%, respectively) for treated *Lmna*^G609G/+^ mice than for untreated *Lmna*^G609G/+^ mice. Notably, ATP synthesis in all the experimental groups was 2-fold higher when mitochondria were assessed in incubation media containing 1 mM magnesium compared to incubation media containing 0.1 mM magnesium, which implies that ATP synthase is simulated by magnesium independently of the effect of treatment.

**Magnesium improves extramitochondrial NADH oxidation**

Oxidation of exogenous NADH by mitochondria in the presence of added extramitochondrial cytochrome c has been described previously (Bodrova *et al*, 1998; Lemeshko, 2000). Oxidation was shown to be insensitive to rotenone, antimycin A, and was suppressed by cyanide (Bodrova *et al*, 1998). The external NADH-cytochrome c reductase electron transport system of the outer membrane of mitochondria is known to have a very high activity (Lemeshko, 2000). Notably, rotenone-insensitive oxidation of external NADH in isolated mitochondria was significantly higher in both treated (4.9-fold) and untreated (4.4-fold) *Lmna*^G609G/+^ and wild-type (4.2-fold) mice in media containing magnesium compared to media in the absence of magnesium (Fig EV4B; Appendix Table S9). This stimulation of NADH oxidation by $Mg^{2+}$ ions was enhancer by addition of exogenous cytochrome c and suppressed by cyanide (Fig EV4B).

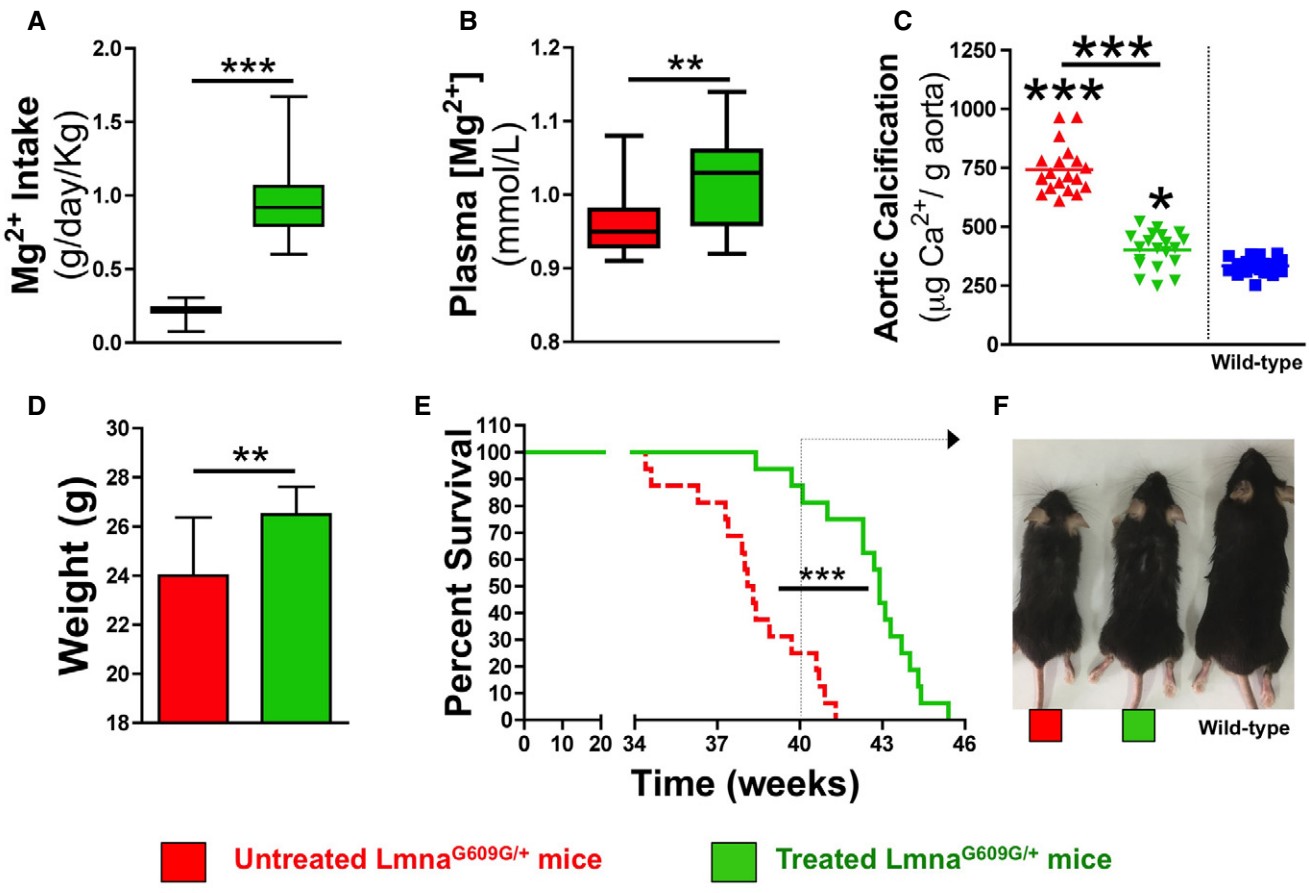

**Figure 4. Oral magnesium treatment improves the longevity of *Lmna*$^{G609G/+}$ mice.**

A, B (A) Magnesium intake and (B) plasma magnesium concentration in 34-week-old untreated and treated *Lmna*$^{G609G/+}$ mice (*n* = 16).
C Calcium content of aortas obtained from 34-week-old wild-type mice and untreated and treated *Lmna*$^{G609G/+}$ mice (*n* = 20).
D Body masses of 34-week-old untreated and treated *Lmna*$^{G609G/+}$ mice.
E Kaplan–Meier graph for untreated and treated *Lmna*$^{G609G/+}$ mice (*n* = 16).
F Representative photographs of 40-wk-old wild-type, untreated, and treated *Lmna*$^{G609G/+}$ mice.

Data information: Results are presented as the mean ± SD. Statistical analyses were performed using Student's *t*-test (A, B, D), log-rank test (E), or one-way ANOVA and Tukey's multiple comparison *post hoc* test (C). **$P < 0.05$, **$P < 0.01$, ***$P < 0.001$.
Source data are available online for this figure.

## Discussion

Magnesium has been shown to effectively prevent mineralization in multiple experimental models of vascular calcification, including cultured VSMCs (Kircelli *et al*, 2012; Louvet *et al*, 2013; Bai *et al*, 2015; Ter Braake *et al*, 2018), uremic rats (Diaz-Tocados *et al*, 2017), and mouse models of pseudoxanthoma elasticum (PXE; *Abcc6*-null mice) (Gorgels *et al*, 2010) and generalized arterial calcification of infancy (GACI; *Enpp1*-null mice) (Kingman *et al*, 2017).

Hydroxyapatite ($Ca_{10}(PO_4)_6(OH)_2$) is the main calcium phosphate crystal found in *in vivo* (Lee *et al*, 2012) and *in vitro* (Villa-Bellosta *et al*, 2011) calcification, and its formation has been shown to be essential for VSMC transdifferentiation (Sage *et al*, 2011; Villa-Bellosta *et al*, 2011; Villa-Bellosta, 2018) and to induce VSMC death (Ewence *et al*, 2008). Although it has been suggested that the incorporation of $Mg^{2+}$ into hydroxyapatite crystals, to form the mineral whitlockite ($Ca_9Mg(HPO_4)(PO_4)_6$), may reduce crystal pathogenicity

by increasing their solubility, previous studies have exclusively identified hydroxyapatite, and not whitlockite, in deposits in calcifying VSMCs supplemented with magnesium (Louvet *et al*, 2015).

While the exact mechanisms whereby magnesium prevents calcification remain to be determined, our data seem to exclude a physicochemical role of magnesium in altering calcium phosphate crystal growth, as evidenced by our finding showing similar calcium deposition in fixed *Lmna*$^{G609G/+}$ VSMCs incubated with high magnesium to those incubated in medium containing a standard concentration of magnesium (Fig 3). Thus, the beneficial role of magnesium in attenuating vascular calcification is likely to be linked to an active cellular role.

Consistent with this, a synergistic effect has been demonstrated when magnesium and ATP are used together in solution to delay the conversion of a slurry of amorphous calcium phosphate to crystalline hydroxyapatite (Blumenthal *et al*, 1977). Moreover, ATP has also been found to prevent vascular calcification by directly

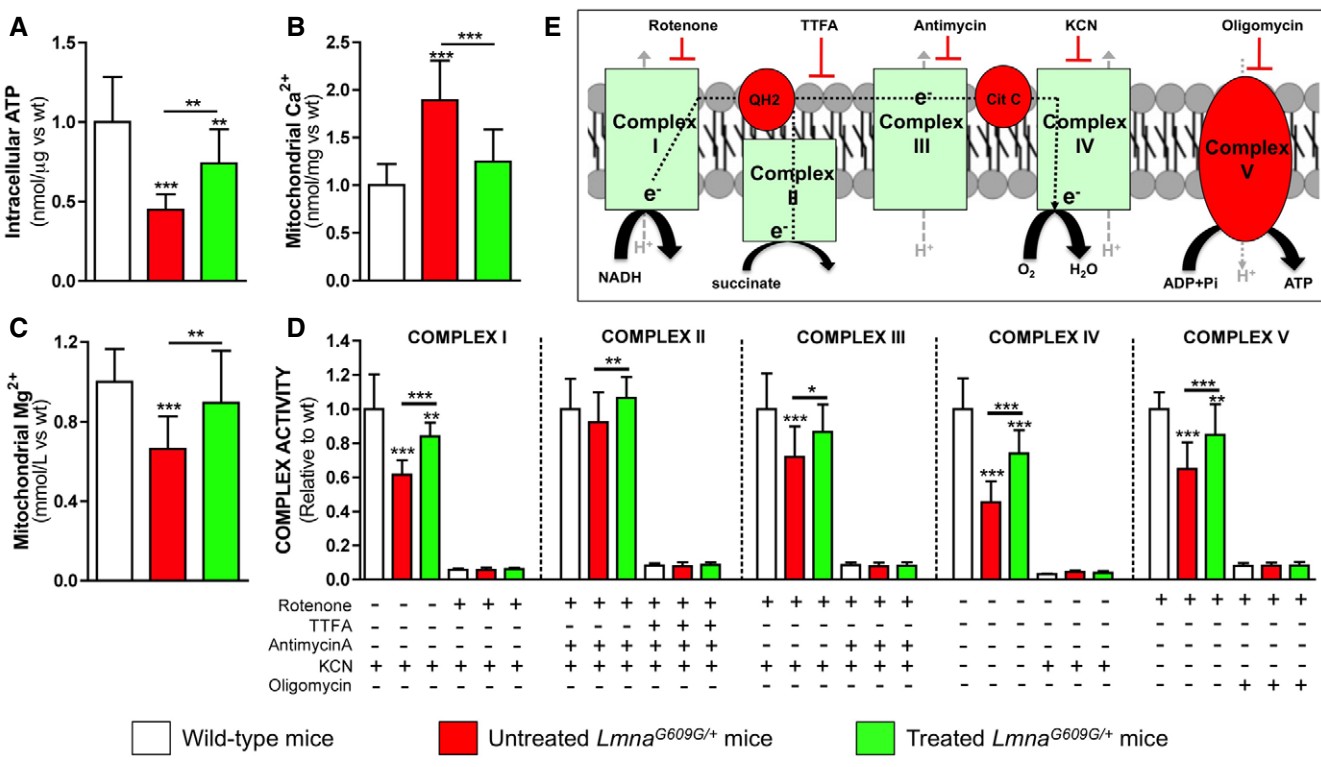

**Figure 5. Magnesium improves mitochondrial ATP synthesis in *Lmna*^G609G/+ mice.**

A  ATP concentration in liver homogenates obtained from 34-week-old wild-type, untreated, or treated *Lmna*^G609G/+ mice.

B  Mitochondrial calcium measured in liver mitochondria isolated from the indicated experimental mouse groups.

C  Magnesium concentration in isolated mitochondria.

D  Activities of the indicated mitochondrial complexes (I, II, III, IV, and V) in the absence or presence of rotenone (2 μM), 2-thenoyltrifluoroacetone (TTFA; 1 mM), antimycin A (10 μM), potassium cyanide (KCN; 1 mM), or oligomycin (10 μM).

E  The boxed scheme shows the five mitochondrial complexes involved in the electron transport chain and their known inhibitors. Data information: Results are presented as mean ± SD ($n = 16$). One-way ANOVA and Tukey's multiple comparisons *post hoc* test were used for statistical analysis. *$P < 0.05$; **$P < 0.01$; ***$P < 0.001$.

Source data are available online for this figure.

---

inhibiting calcium phosphate crystal formation (Villa-Bellosta & Sorribas, 2013). In addition, extracellular ATP is the principal source of extracellular pyrophosphate, a key endogenous inhibitor of calcification (Villa-Bellosta & O'Neill, 2018). We have shown increases in ATP availability, in both treated *Lmna*^G609G/+ VSMCs and mice, which could be explained by increases in the synthesis of ATP in mitochondria and glycolysis.

$Mg^{2+}$ is an important divalent cation in cells that stabilizes nucleic acid and protein structure, and mediates magnesium-dependent enzymatic reactions as a cofactor, including enzymatic reactions involving ATP (Pilchova *et al*, 2017). The mammalian mitochondrial ATP synthase (complex V) catalyzes ATP synthesis from ADP, phosphate, and magnesium using energy generated by an electrochemical gradient of protons produced by the electron transport chain (Chen *et al*, 2006). The mitochondria also generate ROS as a consequence of inefficiencies in the electron transport chain (Murphy, 2009), which cause oxidative stress, DNA damage, and cellular senescence, molecular defects that are found in the premature aging syndrome HGPS (Viteri *et al*, 2010; Gordon *et al*, 2014).

Notably, HGPS fibroblasts generate higher concentrations of ROS than normal fibroblasts (Richards *et al*, 2011). Furthermore, the basal expression of antioxidant enzymes, which defend cells against ROS-induced damage, is also lower in HGPS fibroblasts (Yan *et al*, 1999). Moreover, in HGPS fibroblasts, a marked down-regulation of mitochondrial oxidative phosphorylation proteins, accompanied by severe mitochondrial dysfunction, has been observed, along with a marked reduction in COX activity (cytochrome c oxidase; mitochondrial complex IV) and a significant increase in glycolytic dependency (Rivera-Torres *et al*, 2013; Aliper *et al*, 2015). Therefore, the higher oxidative stress in HGPS cells could be as result of greater ROS formation (see Fig 6), due to defective mitochondrial oxidative phosphorylation, as well as lower ROS-counteracting antioxidative capacity (Kubben *et al*, 2016). Furthermore, emerging evidence suggests that COX dysfunction is invariably associated with greater mitochondrial ROS generation (Srinivasan & Avadhani, 2012; Kadoguchi *et al*, 2020). Moreover, $Ca^{2+}$ accumulation can impair mitochondrial function, leading to lower ATP production and greater release of ROS (Brookes *et al*, 2004; Peng & Jou, 2010; Santulli *et al*, 2015). In

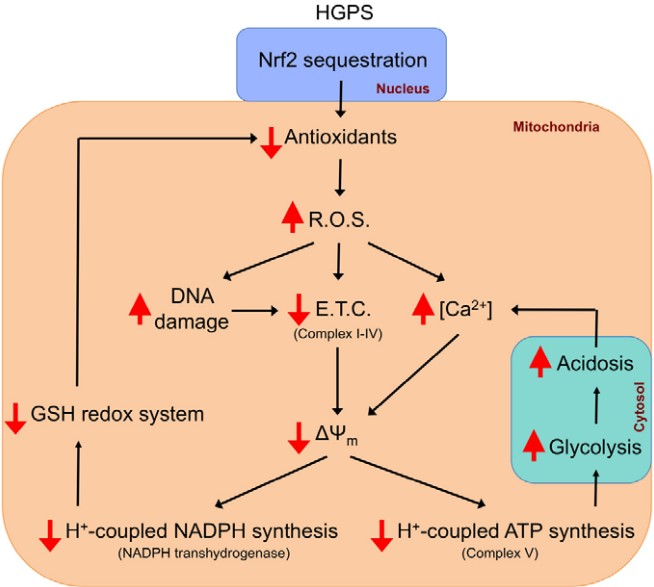

**Figure 6. Proposed model showing the main alterations found in Hutchinson–Gilford progeria syndrome (HGPS).**
R.O.S: reactive oxygen species; E.T.C: electron transport chain. $\Delta\Psi_m$; mitochondrial membrane potential.

which is required for both $H^+$-coupled mitochondrial ATP synthase to generate ATP (Saraste, 1999) and NADPH transhydrogenase to generate NADPH (Rydström, 2006). Furthermore, magnesium increases the activities of mitochondrial dehydrogenases (Fig EV1E), including pyruvate dehydrogenase, isocitrate dehydrogenase, and $\alpha$-ketoglutarate dehydrogenase (Pilchova *et al*, 2017). Therefore, mitochondrial $NADP^+$-dependent isocitrate dehydrogenase could also contribute to the increase in NADPH availability. Moreover, magnesium can reverse the effects of calcium-induced $\Delta\Psi_m$ depolarization (Racay, 2008) and inhibit mitochondrial ROS generation (Kowaltowski *et al*, 1998). Furthermore, several studies (including the present study) have shown that magnesium has a stimulatory effect on the NADH-cytochrome c reductase systems located in the outer mitochondrial membrane (Bodrova *et al*, 1998; Lemeshko, 2000). Therefore, magnesium treatment can also improve exogenous NADH oxidation and the coupled $\Delta\Psi_m$. Consistent with this, the present study has shown improvements in mitochondrial membrane potential and NADPH-coupled glutathione redox system following magnesium treatment. In addition, we have shown that $Lmna^{G609G/+}$ VSMCs and mice treated with high magnesium concentrations have lower ROS concentrations, improvements in both mitochondrial function and mitochondrial ATP synthesis, and thus greater ATP availability, which is necessary for cellular energy supply and survival.

Finally, several studies have shown the beneficial effect of dietary magnesium supplementation in several diseases, including atherosclerosis, diabetes, and heart failure. These studies support the effect of magnesium at the molecular level independently of progerin production and its interactions with nuclear membrane proteins. However, magnesium could interact with nuclear proteins, including telomerase and lamins A, B, and C, which may improve or attenuate its reported beneficial effects. These molecular mechanisms, including other metabolic pathways, signaling pathways, and enzyme activities, will be evaluated in future studies.

addition, $Mg^{2+}$ deficiency is associated with greater production of ROS and the induction of immune and inflammatory reactions (Bussière *et al*, 2002).

Nuclear factor-erythroid 2 p45-related factor 2 (Nrf2) is the primary factor responsible for the protection of cells from oxidative stress, which it does by regulating cytoprotective gene expression, including that of the antioxidant glutathione pathway (Harvey *et al*, 2009). Interestingly, repression of the antioxidant NRF2 pathway has been found in HGPS (Kubben *et al*, 2016). Consistent with this, we have shown significantly lower total glutathione synthesis and GR activity, which were not improved in the presence of a high magnesium concentration. By contrast, both the GSH:GSSG and NADPH:NAD$^+$ ratios, which are indicative of antioxidant status, were improved by magnesium treatment.

According to the redox theory of aging, aging is associated with redox imbalance (Sohal & Orr, 2012; Go & Jones, 2017). Thus, both increases in mitochondrial ROS and a deterioration in antioxidant status stimulate aging (see Fig 6). The reductions in cytoplasmic and mitochondrial NADPH:NADP$^+$ ratio with aging are associated with reductions in the activities of cytoplasmic and mitochondrial GRs (and thioredoxin reductases), which lead to greater oxidization of the glutathione redox couple (GSH:GSSG), resulting in lower activities of glutathione peroxidases (and thioredoxin peroxidases) (Bradshaw, 2019). This oxidation of the NADPH-linked redox systems with aging also causes the oxidation of ascorbate (vitamin C) and tocopherols (vitamin E) (Ren *et al*, 2017). In the present study, we found a significant reduction in the total antioxidant capacity in both $Lmna^{G609G/+}$ cells and mice, which was ameliorated by magnesium treatment.

The uptake of $Ca^{2+}$ ions by mitochondria should depolarize mitochondrial membranes ($\Delta\Psi_m$) (Chalmers & McCarron, 2008),

## Conclusion

Plasma magnesium levels do not always accurately reflect total body magnesium stores. In fact, plasma magnesium levels can be normal despite depletion of the total body magnesium content. However, several studies have shown a connection between magnesium deficiency and aging. In addition, a statistically significant inverse correlation between the level of magnesium in drinking water and cardiovascular mortality has been reported in observational epidemiological studies (Rosique-Esteban *et al*, 2018). Therefore, a lack of magnesium in drinking water and food may underlie the aging-associated progressive deterioration of physiological functions, including the redox balance, senescence, and vascular calcification, while high magnesium intake may delay aging. Consistently, the current study demonstrated that addition of magnesium to drinking water significantly extended longevity in progeroid mice. Therefore, dietary magnesium supplementation may be beneficial in children with HGPS, even those who appear to be normomagnesemic (Merideth *et al*, 2008). Further experiments are needed to test the effect of magnesium supplement in human HGPS context and validate the results obtained in mouse HGPS model.

# Materials and Methods

### Animals

Male $Lmna^{G609G/+}$ and wild-type (C57/BL6) littermates were used at the indicated age. $Lmna^{G609G/+}$ was designed by Carlos López-Otín research group (Oviedo University, Spain) in close collaboration with two French teams (one lead by Nicolas Lévy and the other by Bernard Malissen). The protocol was approved by ethics committees both the FIIS-FJD (Fundación Instituto de Investigación Sanitaria, Fundación Jiménez Díaz) and Madrid Community (PROEX177/15); and conformed to directive 2010/63EU and recommendation 2007/526/EC regarding the protection of animals used for experimental and other scientific purposes, enforced in Spanish law under RD1201/2005. Sample size for animal studies was estimated based on our previous experience and mouse availability. Animals were grouped by genotype. No blinding was performed.

### Aorta isolation

$Lmna^{G609G/+}$ mice were euthanized via carbon dioxide inhalation and thoracic aorta tissue was perfused with saline and removed according to previously published protocol (Villa-Bellosta & Hamczyk, 2015).

### Quantification of aortic calcification

To quantify the calcium content, mice aortas were dried, weighed, and treated with 0.6 M HCl 24 h. Then, calcium was quantified using a colorimetric QuantiChrom Calcium Assay Kit (BioAssay Systems, Hayward, CA).

### Plasma magnesium levels

Blood was collected in heparin-containing tubes and separated into plasma by centrifugation at 5,000 $g$ for 10 min at 4°C. Mg was determined with the QuantiChrom Magnesium Assay Kit (BioAssay System) by the manufacturer's instructions.

### VSMCs isolation and culture

VSMCs were obtained from three independents wild-type or $Lmna^{G609G/+}$ mice aorta pools (8–10 aortas per pool) by double digestion with collagenase method (Villa-Bellosta & Hamczyk, 2015). VSMCs were grown in minimum essential medium Eagle (MEM) supplemented with 1 mM L-glutamine, 100 IU/ml penicillin, 100 μg/ml streptomycin, and 10% fetal bovine serum at 37°C in a humidified atmosphere of 5% $CO_2$. All cell culture reagents were obtained from Invitrogen (Paisley, UK). MEM containing 0.8 mM $Mg^{2+}$ was supplemented with 1 mM $MgCl_2$ (Sigma-Aldrich; final concentration of 1.8 mM) to obtain magnesium-enriched medium. After first trypsinization (passage 1), cells were incubated in MEM containing 0.8 or 1.8 mM $Mg^{2+}$ to passage 8. Cells were grown considering 1:3 splitting during trypsinization. VSMCs were used directly from same passage (8–9), without a quiescent intermediate step. Figure 1B shows the growth rate after the 8[th] trypsinization (passage 8).

### Cell proliferation

The replicative incorporation of 5-bromodeoxyuridine (BrdU) was measured using BrdU Cell Proliferation ELISA Kit (Abcam, ab126556) by the manufacturer's instructions.

### Cell viability

The cell viability was assessed measuring the mitochondrial activity by using tetrazolium salt which in cleavage to formazan by cellular mitochondria dehydrogenase, (WST-1 Assay Kit, ab65475, Abcam). Water-soluble WST-1 was added to each well, and the absorbance was measured using a scanning multiwell microplate according to the manufacturer's protocol.

### β-Gal activity

Senescence-associated β-galactosidase activity was measured in cell lysates by plate reader using a fluorescent probe (β-gal Activity Assay Kit, BioVision), according to the manufacturer's protocols. Beta-galactosidase hydrolyzes a non-fluorescent substrate to generate a strong fluorescent product, which was measured (Ex/Em = 480/520) in two time points (0 and 60 min).

### ATP quantification

ATP was measured by a coupled luciferin/luciferase reaction with an ATP Determination Kit (Invitrogen). Cells were treated with lysis buffer (50 Tris–HCl mM, 150 NaCl mM, 1% Triton X-100 containing inhibitor cocktail, pH 7.4). VSMCs or liver lysates (intracellular ATP) and ATP standards were measured, according to the manufacturer's instructions (Villa-Bellosta, 2019). For mitochondrial ATP measurement, VSMCs were previously incubated with or without oligomycin (10 μM) for 15 min. Mitochondrial ATP was calculated by the subtraction of intracellular ATP levels (with oligomycin) from total ATP (without oligomycin).

### Mitochondrial membrane potential ($\Delta\Psi_m$) measurement

The $\Delta\Psi_m$ was assessed by plate reader using a fluorescent probe (JC-10; ab112134, Abcam) by following the manufacturer's instruction. When mitochondria are polarized electrically, JC-10 forms J-aggregates that emit orange-red fluorescence. J-monomers, indicating depolarized mitochondria, emit green fluorescence. The $\Delta\Psi_m$ was calculated by a ratio of red/green fluorescence, indicating mitochondria depolarization with smaller ratio.

### Oxygen consumption rate

Oxygen consumption rate (OCR) by VSMCs ($5 \times 10^4$ cells/well; Fig 2) or isolated mitochondria (Fig EV4) was measured on standard fluorescence plate reader using the Extracellular $O_2$ Consumption Assay (Abcam, ab197243) according to the manufacturer's protocol. OCR was measured using fluorescence microplate reader (excitation/emission wavelength of 380/650 nm).

Liver mitochondria were isolated by the standard method of homogenization followed by low ($700 \times g$, 10 min) and high ($12,000 \times g$, 15 min) centrifugation using the isolation medium

composed of 250 mM sucrose, 2 mM EGTA, and 5 mM MOPS-KOH (pH 7.4). Measurement buffer contained 60 mM sucrose, 0.5 mM EGTA, 5 mM MOPS-KOH (pH 7.4), and 1 mM NADH.

### ROS detection

Reactive oxygen species (ROS) were measured using the cell permeant reagent 2′,7′-dichlorofluorescin diacetate (DCFDA) according to the manufacturer's protocol (Abcam, ab113851). DCFDA is deacetylated by cellular esterases and oxidized by ROS into a highly fluorescent compound which was measured using a fluorescence microplate reader (excitation/emission wavelength of 488/535 nm). Amplex$^®$ Red reagent (10-acetyl-3,7-dihydroxyphenoxazine) was used to detect hydrogen peroxide ($H_2O_2$). In the presence of peroxidase, the Amplex$^®$ Red reagent reacts with $H_2O_2$ to produce the red-fluorescent oxidation product, resorufin, which was measured using fluorescence microplate reader (excitation/emission wavelength of 430/590 nm), according to the manufacturer's protocol (A22188, Invitrogen). To detect production of mitochondrial superoxide radical in live cells, the Mitochondrial Superoxide Detection Kit (ab219943, Abcam) was used, according to the manufacturer's protocols. Superoxide was measured using fluorescence microplate reader (excitation/emission wavelength of 540/590 nm).

### Total antioxidant capacity

Total antioxidant capacity (TAC) was determined using a commercially available assay kit (Abcam, ab65329) which utilizes the conversion of $Cu^{2+}$ ions to $Cu^+$ through endogenous protein and small molecule antioxidants, standardized to Trolox equivalents. VSMCs and liver were used for analysis of TAC according to the manufacturer's protocols. Colorimetric activity was measured by optimal density at 570 nm.

### NADPH-coupled glutathione redox system assay

Reduced/oxidized glutathione (GSH/GSSG) ratio, NADPH/NADP ratio, and glutathione reductase (GR) activity in both VSMCs and liver lysates were measured with commercials kit (Abcam, ab138881, ab176724, and ab83461, respectively), using a 96-well plate reader, according to the manufacturer's protocols. Liver samples were measured in triplicate on the same plate, and fluorescence/colorimetric values were normalized to micrograms of protein loaded in the assay per sample. Protein was measured with the Pierce BCA Protein Assay Kit (Thermo Scientific, Rockford, USA), according to the manufacturer's protocols.

GSH and total glutathione were determined by changes in fluorescence intensity (excitation/emission wavelength of 490/520 nm). 20 mg of liver tissue or VSMCs ($5 \times 10^6$) were homogenized/lysed in 400/100 μl cold lysis buffer, respectively. Homogenates/lysates were centrifuged at top speed for 15 min at 4°C. Supernatants were deproteinized using Deproteinizing Sample Preparation Kit-TCA (Abcam, ab204708) according to the manufacturer's protocol.

NADPH and total nicotinamide adenine dinucleotide phosphate ($NADPH^+NADP^+$) were determined by changes in fluorescence intensity (excitation/emission wavelength of 540/590 nm). 20 mg of liver tissue or VSMCs ($5 \times 10^6$) were homogenized/lysed in 400/

100 μl lysis buffer, respectively. Homogenates/lysates were centrifuged at 2,500/1,500 rpm, respectively, for 5 min at RT.

GR activity was measured by optimal density at 405 nm. 20 mg of liver tissue or $1 \times 10^6$ VSMCs were homogenized/lysed in 200 μl cold assay buffer and centrifuged at $10,000 \times g$ for 15 min at 4°C. Supernatant was pre-treated to destroy GSH before the assay, according to the manufacturer's protocol.

### Mitochondrial and cytosolic ATP synthesis

Liver mitochondria were isolated by the standard procedure of differential centrifugation using the isolation medium composed of 250 mM sucrose, 2 mM EGTA, and 5 mM MOPS-KOH (pH 7.4). Mitochondria were washed and finely suspended in the medium composed of 120 mM KCl, 20 mM MOPS, and 0.5 mM EGTA (KME medium). Mitochondria protein was measured with the Pierce BCA Protein Assay Kit (Thermo Scientific, Rockford, USA), according to the manufacturer's protocols. Mitochondria (1 mg of protein/ml) were incubated in KME medium containing 5 mM succinate, 2 μM rotenone, 10 mM NaCl, 800 pmol of A23187/mg of protein, and 5 mM phosphate (10 μCi/ml of $^{32}Pi$ as radiotracer) at 37°C. After 5 min, 1.2 mM ADP was added, and the reaction was stopped 30 s later by addition of 200 μl of 30% (w/v) cold trichloroacetic acid. Orthophosphate ($^{32}Pi$) was separated from ATP ($[\gamma^{32}P]ATP$) by molybdate method, as previously described (Villa-Bellosta, 2018, 2019). Briefly, 20 μl of sample was mixed with 400 μL of ammonium molybdate (to bind the orthophosphate) and 0.75 M sulfuric acid. Samples were then extracted with 800 μl of isobutanol/petroleum ether (4:1) to separate the phosphomolybdate from pyrophosphate and ATP. Next, 400 μl of the aqueous phase containing ATP was removed and subjected to radioactivity counting (Tri-Carb 2810TR, PerkinElmer).

For cytosolic ATP synthesis, VSMCs ($1 \times 10^6$) were incubated in MEM containing 10 μM oligomycin and 10 μCi/ml of phosphate-32 ($^{32}Pi$) as radiotracer, at 37°C. The reaction was stopped by addition of 200 μl of 30% (w/v) cold trichloroacetic acid. Orthophosphate ($^{32}Pi$) was separated from ATP ($[\gamma^{32}P]ATP$) by molybdate method.

### Glycolysis assay

Extracellular acidification and lactate were measured with fluorometric kits (Abcam, ab197244 and ab169557, respectively), using a 96-well plate reader, according to the manufacturer's protocols. Extracellular acidification was determined in VSMCs ($5 \times 10^5$ cells/well) by changes in fluorescence intensity (excitation/emission wavelength of 380/615 nm), using a water-soluble and cell-impermeable pH-sensitive reagent. L-lactate was determined by changes in fluorescence intensity (excitation/emission wavelength of 535/587 nm). VSMCs ($1 \times 10^6$) were homogenized with 110 μl cold lactate assay buffer on ice and centrifuged at 14,000 $g$ for 5 min. Supernatant was measured in duplicate on the same plate.

### Mitochondrial calcium and magnesium

For calcium accumulation by mitochondria, VSMCs were incubated in MEM containing 10 μCi/ml calcium-45 ($^{45}Ca^{2+}$) as a radiotracer. After 24 h, VSMCs were washed five times in MEM. Mitochondria were isolated from VSMCs by method of homogenization followed

by low- and high-speed centrifugation at 4°C. The homogenate was centrifuged at $1,000 \times g$ for 10 min. Mitochondria were sedimented at $3,500 \times g$ for 15 min. The pellet contains the isolated mitochondria was washed and centrifuged at $12,000 \times g$ for 15 min. The isolation medium contained 250 mM sucrose, 0.5 mM EGTA, and 5 mM MOPS-KOH (pH 7.4). EGTA was excluded from washing medium. Mitochondria pellets was resuspended in liquid scintillation counting (UltraGold, 6013329, PerkinElmer) and subjected to radioactivity counting (Tri-Carb 2810TR, PerkinElmer).

Liver mitochondria were isolated by the method of homogenization followed by low ($1,000 \times g$, 10 min)- and high ($3,500 \times g$, 15 min)-speed centrifugation at 4°C. Mitochondria were washed and finely suspended in $dH_2O$. Mitochondria protein was measured with the Pierce BCA Protein Assay Kit (Thermo Scientific, Rockford, USA), according to the manufacturer's protocols. Calcium was measured using a colorimetric QuantiChrom Calcium Assay Kit (BioAssay System, Hayward, CA) according to the manufacturer's protocols.

Mitochondrial magnesium concentration was measured in isolated mitochondria using mag-fura 2-AM (Thermo Fisher) as described previously (Kolisek *et al*, 2003). Briefly, mitochondria were loaded with 5 μM mag-fura 2-AM for 40 min at 25°C and washed twice to remove excess dye. Magnesium concentration was determined by measuring the fluorescence of the probe-loaded mitochondria with excitation at 340 and 380 nm, and emission at 510 nm. Mitochondrial magnesium was calculated from the 340/380 nm ratio according to the formula of Grynkiewicz *et al* (Grynkiewicz *et al*, 1985). The minimum ($R_{min}$) and maximum ($R_{max}$) ratios were obtained at the end of each experiment. $R_{max}$ was obtained by the addition of SDS (10% w/v) and $MgCl_2$ (25 mM). $R_{min}$ was detected by addition of EDTA (50 mM, pH 8).

### Calcification assay and quantification

VSMCs were used from same passage (8 or 9), considering 1:3 splitting during trypsinization. Cells were grown to confluence and used after a quiescent intermediate step (overnight in culture media containing 0.1% fetal bovine serum). Calcification assays were performed on cells incubated for 7 days in MEM supplemented with 1 mM L-glutamine, 100 IU/ml penicillin, 100 μg/ml streptomycin, 0.1% fetal bovine serum, and 2 mM phosphate (phosphate-calcifying medium), as described previously (Villa-Bellosta, 2018; Villa-Bellosta *et al*, 2011, p.; Villa-Bellosta & Hamczyk, 2015). Phosphate-calcifying medium was replaced every day. To quantify the calcium content of VSMC, wells were treated with 0.6 M HCl overnight at 4°C and analyzed using a colorimetric QuantiChrom Calcium Assay Kit (BioAssay System, Hayward, CA). Phosphate, magnesium chloride, phosphonoformic acid, and pyrophosphate were obtained from Sigma-Aldrich. Cells were fixed as described previously (Villa-Bellosta & Sorribas, 2009).

### Mitochondrial complex I-V activities

Mitochondria were isolated from mouse liver tissues using Mitochondria Isolation Kit for tissue (Cayman Chemical, #701010). The activities of electron transport chain complexes I, II, III, IV, and V were determined using complex I activity assay kit (Cayman Chemical, #700930), complex II activity assay kit (Cayman Chemical,

**The paper explained**

**Problem**

Loss of antioxidant capacity, excessive generation of reactive oxygen species (ROS) and mitochondrial dysfunction contribute to the main symptoms observed in premature aging associated to Hutchinson-Gilford progeria syndrome (HGPS).

**Results**

Here, we show that treatment with exogenous magnesium improved the mitochondrial function and reduced oxidative stress both in HGPS mice and vascular smooth muscle cells. Magnesium treatment improved mitochondrial ATP synthesis, and thus greater ATP availability, which is necessary for cellular energy supply and survival. Consistently, magnesium treatment improved mice longevity and reduced vascular calcification.

**Impact**

This study shows antioxidant properties of magnesium and its capacity to increase the ATP viability in a mouse model of HGPS, which in turn suggest novel possibilities for treating children with HGPS.

#700940), complex II/III activity assay kit (Cayman Chemical, #700950), complex IV activity assay kit (Cayman Chemical, #700990), and complex V activity assay kit (Cayman Chemical, #701000), respectively, following the manufacturer's instruction. Rotenone, 2-thenoyltrifluoroacetone (TTFA), antimycin A, potassium cyanide (KCN), and oligomycin were obtained from Sigma-Aldrich.

### Magnesium treatment

To assess the effect of supplemental magnesium on $Lmna^{G609G/+}$ mice, their drinking water (containing 39 mg/l magnesium) was supplemented with 15 g/l $MgCl_2$. Thereafter, the consumption of food (containing 0.17% magnesium) and that of both untreated and treated water was measured in the mice between 8 and 34 weeks of age. Magnesium intake was measured twice a week. Average daily food and water intake was calculated per day and by weight of the mouse in each cage.

### Statistical analyses

Results are presented as means $\pm$ SD. The Kolmogorov–Smirnov test was used to assess the normality of the data. Student's *t*-test or one-way ANOVA and Tukey's multiple comparison posttest were used for statistical analysis. Asterisks near the top of the columns compare untreated or treated cells/mice with respect to control (wild type). Longevity was assessed by the Kaplan–Meier methods. All statistical analyses were performed using GraphPad Prism 5 software. Differences were considered significant at $P < 0.05$. Randomization or blinding was not applicable in this study.

## Data availability

All source data of this study are available in the supplementary material of the article. Other data that support the findings of

this study are available from the corresponding authors upon request.

**Expanded View** for this article is available online.

## Acknowledgements
We thank Daniel Azpiazu, Ana Lobo, Ana de la Calle, Veronica, and Jose Luis for excellent technical assistance. We also thank the group of Prof. Carlos López-Otín for the generous gift of the HGPS mouse model and the personnel of FIIS-FJD Animal Facility for animals' production and maintenance. This study was supported by grants from Progeria Research Foundation (PRF-2016-68) from USA and Spanish Ministerio de Economía y Competitividad (SAF-2014-60669-JIN).

## Author contribution
RV-B conceived and designed the study, conducted the experiments, acquired the data, analyzed and interpreted the data, provided the reagents, and wrote the manuscript.

## Conflict of interest
The author declares that he has no conflict of interest.

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
