## [Review Process File · EMBO Molecular Medicine]

Dietary magnesium supplementation improves lifespan in a mouse model of progeria

Ricardo Villa-Bellosta

DOI: [10.15252/emmm.202012423](https://doi.org/10.15252/emmm.202012423)

Corresponding author: Ricardo Villa-Bellosta (clubcodigogenetico@gmail.com)

Review Timeline:

Submission Date:	29th Mar 20
Editorial Decision:	21st Apr 20
Revision Received:	11th Jun 20
Editorial Decision:	7th Jul 20
Revision Received:	28th Jul 20
Accepted:	28th Jul 20

Editor: Jingyi Hou

Transaction Report:

21st Apr 2020

Dear Dr. Villa-Bellostá,

Thank you for the submission of your manuscript to EMBO Molecular Medicine. We have now received feedback from the three referees who agreed to evaluate your manuscript. As you will see from the reports below, the referees acknowledge the potential interest of the study. However, they also raise substantial concerns about your work, which should be convincingly addressed in a major revision of the present manuscript. In particular, both referee #1 and #2 commented on the use of heterozygous G609G/+ mice instead of homozygous G609G/G609G mice, and referee #1 is also concerned about the clinical relevance of the presented findings for treating human patients due to the high dose of Mg⁺⁺ used in these experiments. These concerns must be satisfactorily addressed. Further, additional experiments and analyses (especially imaging data as requested by referee #2 and #3) are required to strengthen the conclusion.

We would welcome the submission of a revised version within three months for further consideration. Please note that EMBO Molecular Medicine strongly supports a single round of revision and that, as acceptance or rejection of the manuscript will depend on another round of review, your responses should be as complete as possible.

We are aware that many laboratories cannot function at full efficiency during the current COVID-19/SARS-CoV-2 pandemic and have therefore extended our "scooping protection policy" to cover the period required for a full revision to address the experimental issues. Please let me know should you need additional time, and also if you see a paper with related content published elsewhere.

I look forward to receiving your revised manuscript.

Yours sincerely,

Jingyi Hou

Jingyi Hou
Editor
EMBO Molecular Medicine

*** Instructions to submit your revised manuscript ***

**** PLEASE NOTE **** As part of the EMBO Publications transparent editorial process initiative (see our Editorial at <https://www.embopress.org/doi/pdf/10.1002/emmm.201000094>), EMBO Molecular Medicine will publish online a Review Process File to accompany accepted manuscripts.

To submit your manuscript, please follow this link:

Link Not Available

- 1) a .doc formatted version of the manuscript text (including Figure legends and tables). Please make sure that the changes are highlighted to be clearly visible to referees and editors alike.
- 2) separate figure files*
- 3) supplemental information as Expanded View and/or Appendix. Please carefully check the authors guidelines for formatting Expanded view and Appendix figures and tables at <https://www.embopress.org/page/journal/17574684/authorguide#expandedview>
- 4) a letter INCLUDING the reviewers' reports and your detailed responses to their comments (as Word file)

Also, and to save some time should your paper be accepted, please read below for additional information regarding some features of our research articles:

- 5) The paper explained: EMBO Molecular Medicine articles are accompanied by a summary of the articles to emphasize the major findings in the paper and their medical implications for the non-specialist reader. Please provide a draft summary of your article highlighting
 - the medical issue you are addressing,
 - the results obtained and
 - their clinical impact.

6) For more information: There is space at the end of each article to list relevant web links for further consultation by our readers. Could you identify some relevant ones and provide such information as well? Some examples are patient associations, relevant databases, OMIM/proteins/genes links, author's websites, etc...

7) Author contributions: the contribution of every author must be detailed in a separate section (before the acknowledgments).

8) EMBO Molecular Medicine now requires a complete author checklist (<https://www.embopress.org/page/journal/17574684/authorguide>) to be submitted with all revised manuscripts. Please use the checklist as a guideline for the sort of information we need WITHIN the manuscript as well as in the checklist. This is particularly important for animal reporting, antibody dilutions (missing) and exact p-values and n that should be indicated instead of a range.

9) Every published paper now includes a 'Synopsis' to further enhance discoverability. Synopses are displayed on the journal webpage and are freely accessible to all readers. They include a short stand first (maximum of 300 characters, including space) as well as 2-5 one sentence bullet points that summarise the paper. Please write the bullet points to summarise the key NEW findings. They should be designed to be complementary to the abstract - i.e. not repeat the same text. We encourage inclusion of key acronyms and quantitative information (maximum of 30 words / bullet point). Please use the passive voice. Please attach these in a separate file or send them by email, we will incorporate them accordingly.

You are also welcome to suggest a striking image or visual abstract to illustrate your article. If you do please provide a jpeg file 550 px-wide x 400-px high.

10) A Conflict of Interest statement should be provided in the main text

11) Please note that we now mandate that all corresponding authors list an ORCID digital identifier. This takes <90 seconds to complete. We encourage all authors to supply an ORCID identifier, which will be linked to their name for unambiguous name identification.

Currently, our records indicate that the ORCID for your account is 0000-0002-1680-552X.

Link Not Available

12) The system will prompt you to fill in your funding and payment information. This will allow Wiley to send you a quote for the article processing charge (APC) in case of acceptance. This quote takes into account any reduction or fee waivers that you may be eligible for. Authors do not need to pay any fees before their manuscript is accepted and transferred to our publisher.

Photos 400-800 DPI

*Additional important information regarding figures and illustrations can be found at <http://bit.ly/EMBOPressFigurePreparationGuideline>

***** Reviewer's comments *****

Referee #1 (Comments on Novelty/Model System for Author):

The right mice model, the right experiments whose results support the conclusion.
Pertinent biological data, but not applicable to human patients

Referee #1 (Remarks for Author):

The manuscript by Ricardo Villa-Belostá, entitled « Dietary magnesium supplementation improves lifespan in a mouse model of progeria » investigates a new and original aspect of this accelerated and premature aging disease, namely magnesium cellular functions, using the heterozygous G609G/+ mouse model designed by Carlos-Otin research team (Oviedo, Spain) in close collaboration with two french teams (one lead by Nicolas Lévy and the other by Bernard Malissen in Marseille), a precision that could be done in Material and Methods section.

The manuscript explores cultured VSMC from progeria mice model, the mechanism of VSMC calcification in vitro, aorta calcification, progeria mice life span and body weight after Mg⁺⁺ dietary supplementation and several aspects of mitochondria biology from cultured VSMC or in isolated from mice liver.

The main results of the manuscript, supported by a large number of experimental methods are as following :

1. Mg⁺⁺ increases the viability of cultured VSMC from progeria mice.
2. Mg⁺⁺ enhances ATP production by cultured VSMC, reduces their oxidative stress, reduces mitochondrial Ca⁺⁺ overload induced by lactate and extracellular acidification.
3. In progeria mice liver cells, Mg⁺⁺ improves anti-oxydant defences, increases ATP production by isolated mitochondria and extramitochondrial NADH oxidation.
4. Going closer to progeria pathophysiological mechanism targeting blood vessel wall and leading to HGPS patient death, the authors demonstrated that Mg⁺⁺ reverses both VSMC calcification in vitro, as well as aorta calcification from G609G mice, two events that could explain the mice lifespan increase.

Main comments :

1. The Authors explored the heterozygous G609G/+ mice. Could the same results be obtained using the homozygous G609G/G609G mice ?
2. Details concerning the Mg⁺⁺ supplementation in drinking water of mice appears only in page 9 of Results section : 39 mg/L of Mg⁺⁺ in drinking water supplemented with 15 g/L of MgCl₂.
 - These data have to be given in the first paragraph of the Material & Methods section
 - The reviewer cannot reproduce and cannot understand the two values of Mg⁺⁺ daily intake by HGPS mice, 976 mg/day/kg or 214 mg/day/kg depending on the Mg⁺⁺ diet.

- Extrapolating to human body (70 kg), the equivalent Mg⁺⁺ diet would reach 68 g/day and 15 g/day, i.e. 170 or 40 times higher respectively than the « standard » Mg⁺⁺ diet recommended for the treatment of hypomagnesemia patients, 420 mg/day (table 2, page 60 in Ahmed et al., 2019, Med Sci (Basel) 7(4): 56-63 ; see also : <https://www.ncbi.nlm.nih.gov/books/NBK109825/>).

- Do the Authors confirm the value 15 gram/L of MgCl₂ ?

- The administration to HGPS mice of such an amount of MgCl₂ for 28 weeks could represent a Mg⁺⁺ overload that could result in several adverse effects (de Baaij et al. 2015. Physiological Reviews 95(1): 1-46). A comment by Authors is required, as well as a paragraph in Discussion section.

3. The manuscript will benefit from a discussion regarding Mg⁺⁺ plasma level in normal subjects, in HGPS patients, in WT and G609G mice, and changes during supplementation or aging, even if Mg⁺⁺ plasma level represent only less than 1% of the total body Mg⁺⁺ and that variations in Mg⁺⁺ plasma levels do not reflect changes in intracellular Mg⁺⁺. For example, Lonafarnib induced a decrease in Mg⁺⁺ plasma level in HGPS patients, human aging has been also associated with a decrease in Mg⁺⁺ plasma level.

4. Data from VSMC mitochondria and from isolated liver mitochondria clearly demonstrated that Mg⁺⁺ improves mitochondria functional parameters. However, the manuscript lacks the direct evidence that Mg⁺⁺ enters into mitochondria. The reviewer wonders why the Authors did not quantify mitochondrial Mg⁺⁺ using specific fluorescent probes, as done by Yamanaka et al. 2016. Sci Rep 6: 30027.

5. Besides its several activities in mitochondria respiration, ATP production, oxidative stress..., Mg⁺⁺ is known to interact with lamins A, B and probably progerin (the three proteins bearing an Ig Fold interacting with Mg⁺⁺), with telomerase, all nuclear proteins known to contribute to progeria pathophysiological mechanisms. These aspects could be reported in the Discussion section.

Other questions/comments :

1. Despite some discrepancies between text and figures or between figure and legend (see below), the referee acknowledges the quality of figures as well as the pictures dissecting for the reader mechanisms, inhibitors...

2. Figure 1 :

- 1A : the black curve concerns WT VSMC. Perhaps this legend could be added above the curve. The measurements begin at probably 5 days (?). Do 5 days for VSMC in culture is equivalent to passage 8 or 9 ?

- 1C to 1D : data from VSMC at 30 days in culture/passage 8 or 9 as written in the Material and Methods section ?

- 1D (proliferation) and 1E (ATP) : a discrepancy with text page 5, lines 19 and 21 ; and page 16, line 12.

- Perhaps to precise that asterisks close to the top of columns compare WT and untreated or treated cells

3. Figures 2, 3,4 : data from VSMC at 30 days in culture/passage 8 or 9 ?

4. Figure 4 :

- In manuscript, page 8, line 18, mitochondrial Ca⁺⁺ refers to figure 4C and not 4D

5. Figure 5 : the experimental protocol has to be better described in the Material & Methods section.

6. Figure 6 :

- 6B : what is the Mg⁺⁺ level in plasma from WT mice of the same genetic background and matched in sex and age ?

- 6D ; another discrepancy between text and figures : median survival time from 38.2 to 42.9 weeks in the text and more than 42 weeks and less than 48 weeks in the figure (n = 16 mice). These data have to be homogeneized in text and figure.

- 6E : a comment (in Results or Discussion sections ?) regarding the marked decrease in body

weight after 30 weeks ? Why mice escape from Mg⁺⁺ treatment ?

7. Figure 9 :

- 9B : another bug in 9B : legend quotes « Extramitochondrial NADH oxidation », whereas the Y axis of figure 9B indicates « Oxygen consumption ».
- 9C : text page 12 refers to Figure 9C missing in the corresponding figure.

In conclusion, this manuscript provided some convincing informations regarding the effects of Mg⁺⁺ in cultured cells and organs from G609G progeria mice model.

However, the referee wonders if some results are not related to the very high quantity of Mg⁺⁺ intake by mice.

Moreover, the manuscript final sentence telling that Mg⁺⁺ dietary supplementation could benefit to HGPS children seems to the reviewer far from the clinical reality, at least because the too large quantity of Mg⁺⁺ was administered to mice in these experiments.

Finally, are missing, at least in the Discussion section, several data regarding Mg⁺⁺ plasma levels, dietary Mg⁺⁺ in (aged) man versus HGPS patients, in WT versus progeria mice, interaction of Mg⁺⁺ with lamins (a lack quite surprising knowing the main role of lamins in progeria) and their consequences.

For all these reasons related to its content, and because it has to be carefully improved in its form, the present manuscript is not suitable for publication in EMBO Molecular Medicine.

Referee #2 (Comments on Novelty/Model System for Author):

The model organism used in this study is possibly the best available mouse model of HGPS. In the heterozygous state (used in the submitted research), it also reproduces the heterozygous condition of human disease. I would prefer that the authors report the reason why they chose heterozygous rather than homozygous mice to perform their research.

Referee #2 (Remarks for Author):

The manuscript by Dr. Villa-Belosta reports that Magnesium supplementation can improve the phenotype of progeroid vascular smooth muscle cells (VSMC) and extend lifespan in progeroid LmnaG609G/+ mice. In particular, the author reports that magnesium treatment of VSMC improves ATP synthesis, reduces lactate accumulation and subsequent mitochondrial Calcium overload, reduces oxidative stress and increases cell viability.

Finally, the author shows that oral magnesium supplementation reduces in vivo VSMC calcification and increases lifespan of LmnaG609G/+ mice.

The background and rationale of the study are strong and based on previous results obtained by the author's team and other research groups. Those results have been published in high impact journals.

Major concerns:

- All differences reported in the manuscript, though statistically significant, appear minimal in absolute values. Can the author state that they significantly impact on cellular senescence and organism ageing? Beta-Gal staining of cells and some pictures showing the animal phenotype could give a better idea of the effect of Magnesium supplementation.
- Several biochemical pathways are well described in the manuscript and suggest potential pathogenetic mechanisms. Could the author hypothesize (and test, if possible) how Lmna G609G mutation elicits the biochemical effects reported in the manuscript?

Referee #3 (Remarks for Author):

General comments on EMM-2020-12423 manuscript entitled: "Dietary magnesium supplementation improves lifespan in a mouse model of progeria "

Ricardo Villa-Bellosta investigates the impact of magnesium supplementation on vascular smooth muscle cells (VSMCs) and progeria (LmnaG609G/+) mice mitochondrial function, calcium deposits and lifespan. Using a thorough analysis of several mitochondrial parameters including ATP synthesis, ROS levels, redox status, MMP and energy metabolism, the author provides evidence that magnesium treatment improves all these functions in both in vitro and in vivo studies.

Specific comments

Overall, the study is well designed, the details of the experiments and methodologies are clear and sufficient.

Results

-Paragraph: "Magnesium improves LmnaG609G/+ vascular smooth muscle cell (VSMC) viability"
Figure 1A shows the growth rate of LmnaG609G/+ VSMCs during a period of 30 days. A detailed description of this long-term culture is needed. For instance, how many passages were performed for each treatment condition during this 30-day period?

Hence, morphological images of the cell cultures at early and late time points should be included to visualize some potential changes. Moreover, the growth rate of mock-treated progeria cultures decreases more rapidly than control cells. Upon treatment with magnesium-supplement, the rate of growth increases in progeria VSCM cultures. How about normal VSCM cultures? Is the growth rate also ameliorated upon magnesium treatment? If yes this could suggest that magnesium supplement not only ameliorate progeria cells but possibly normal cells as well.

Paragraph: "Magnesium prevents phosphate-induced LmnaG609G/+ VSMC calcification"
Figure 5: VSMC microscopy images are missing to appreciate the levels of calcification and the morphological changes occurring in progeria VSMC cells by comparison to wild-type cells.

Sentence in conclusion of the discussion section:" Moreover, several studies report an association between cardiovascular disease and the hardness of drinking water due to its differing magnesium content."

Please, rephrase this sentence. I don't understand the meaning.

Beside the above comments, western blot analyses of the lamin A/C status are missing for both in vitro and in vivo studies. What is the impact of magnesium supplement on Lamin A/C levels?

In closing, this is an interesting study. Hence, if magnesium supplementation is sufficient to restore mitochondrial function as indicated by the findings from this report, this opens a new perspective for treatment of HGPS patients and other conditions developing vascular disease.

***** Reviewer's comments *****

Referee #1 (Comments on Novelty/Model System for Author):

The right mice model, the right experiments whose results support the conclusion.
Pertinent biological data, but not applicable to human patients

Referee #1 (Remarks for Author):

The manuscript by Ricardo Villa-Bellosta, entitled « Dietary magnesium supplementation improves lifespan in a mouse model of progeria » investigates a new and original aspect of this accelerated and premature aging disease, namely magnesium cellular functions, using the heterozygous G609G/+ mouse model designed by Carlos-Otin research team (Oviedo, Spain) in close collaboration with two french teams (one lead by Nicolas Lévy and the other by Bernard Malissen in Marseille), a precision that could be done in Material and Methods section.

RESPONSE: Thanks you for finding this loss of information. An additional sentence has been added in Material and Methods section and Acknowledgments.

The manuscript explores cultured VSMC from progeria mice model, the mechanism of VSMC calcification in vitro, aorta calcification, progeria mice life span and body weight after Mg⁺⁺ dietary supplementation and several aspects of mitochondria biology from cultured VSMC or in isolated from mice liver.

The main results of the manuscript, supported by a large number of experimental methods are as following :

1. Mg⁺⁺ increases the viability of cultured VSMC from progeria mice.
2. Mg⁺⁺ enhances ATP production by cultured VSMC, reduces their oxidative stress, reduces mitochondrial Ca⁺⁺ overload induced by lactate and extracellular acidification.
3. In progeria mice liver cells, Mg⁺⁺ improves anti-oxydant defences, increases ATP production by isolated mitochondria and extramitochondrial NADH oxidation.
4. Going closer to progeria pathophysiological mechanism targeting blood vessel wall and leading to HGPS patient death, the authors demonstrated that Mg⁺⁺ reverses both VSMC calcification in vitro, as well as aorta calcification from G609G mice, two events that could explain the mice lifespan increase.

Main comments :

1. The Authors explored the heterozygous G609G/+ mice. Could the same results be obtained using the homozygous G609G/G609G mice ?

RESPONSE: Heterozygous mice were used for two main reasons: 1) HGPS children are heterozygous, and 2) vascular calcification is better observed in heterozygous mice than in homozygous mice because the former mice live longer and thus there is more time for accumulate calcium. Moreover, homozygous mice consume dry food poorly, and their food is usually moistened with water so they can eat it properly. This makes it more difficult to control their magnesium intake. Although the same beneficial effects

of magnesium can be observed in homozygous mice, magnesium should be delivered via daily injection rather than in food.

2. Details concerning the Mg⁺⁺ supplementation in drinking water of mice appears only in page 9 of Results section : 39 mg/L of Mg⁺⁺ in drinking water supplemented with 15 g/L of MgCl₂.

- These data have to be given in the first paragraph of the Material & Methods section

RESPONSE: Details concerning to magnesium supplementation in drinking water was transferred to the Material and Methods section.

- The reviewer cannot reproduce and cannot understand the two values of Mg⁺⁺ daily intake by HGPS mice, 976 mg/day/kg or 214 mg/day/kg depending on the Mg⁺⁺ diet.

RESPONSE: Magnesium intake was measured twice a week. Median daily food and water intake was calculated per day and by weight of the mouse in each cage. For example, using as reference a food and water consumption of 3.5 g/day/mouse and 4 mL/day/mouse, respectively, the consumption of magnesium would be as follows. In both experimental groups, the magnesium intake through food is 5.95 mg/mouse (0.17% magnesium). Therefore, the magnesium consumption is 238 mg/day/Kg for a mouse of 25 g). In the case of drinking water, it was supplemented with 15 g/L of MgCl₂, that is 3,83 g/L Mg²⁺ (MW of Mg²⁺ = 24.3 g/mol; MW of MgCl₂ = 95,21 g/mol). Therefore, the magnesium consumption through water was 15,3 mg/mouse (612 mg/day/Kg mouse, for a mice of 25 g). The total magnesium intake in treated and untreated mouse is 850 (238+612) and 238 mg/day/Kg, respectively, (under these hypothetical data). However, the mean intake represents measures from 8 to 34 weeks. Additional information has been included in Material and Methods section.

- Extrapolating to human body (70 kg), the equivalent Mg⁺⁺ diet would reach 68 g/day and 15 g/day, i.e. 170 or 40 times higher respectively than the « standard » Mg⁺⁺ diet recommended for the treatment of hypomagnesemia patients, 420 mg/day (table 2, page 60 in Ahmed et al., 2019, Med Sci (Basel) 7(4): 56-63 ; see also : <https://www.ncbi.nlm.nih.gov/books/NBK109825/>).

RESPONSE: Due to its unfavorable surface/volume ratio and high metabolic rate, mice ingest much more water and food than humans. Therefore, it is not correct to compare the nutritional requirements of a mouse with those of a human. For example, the food and water consumption by mouse is 10-30% and 10-40%, respectively, of the body weight (PMID: 12467341). This means that a 70kg human should eat 7 kg of food/water per day (for 10%). The important fact of our study is that there is a 4.6-fold difference in magnesium intake in treated mice. In humans, the impact of the highest allowed consumption of magnesium should be analyzed.

- Do the Authors confirm the value 15 gram/L of MgCl₂ ?

RESPONSE: The authors confirm 15 g/L of MgCl₂.

• The administration to HGPS mice of such an amount of MgCl₂ for 28 weeks could represent a Mg⁺⁺ overload that could result in several adverse effects (de Baaij et al. 2015. Physiological Reviews 95(1): 1-46). A comment by Authors is required, as well as a paragraph in Discussion section.

RESPONSE: 50 g/L of MgCl₂ or MgSO₄ have been used in several studies showing beneficial effect without reporting significant adverse effects (for example, see the following studies: PMID: 11348887 and PMID: 30626750). Moreover, 15 g/L of MgCl₂ represent 3.8 g/L magnesium (Mg²⁺).

3. The manuscript will benefit from a discussion regarding Mg⁺⁺ plasma level in normal subjects, in HGPS patients, in WT and G609G mice, and changes during supplementation or aging, even if Mg⁺⁺ plasma level represent only less than 1% of the total body Mg⁺⁺ and that variations in Mg⁺⁺ plasma levels do not reflect changes in intracellular Mg⁺⁺. For example, Lonafarnib induced a decrease in Mg⁺⁺ plasma level in HGPS patients, human aging has been also associated with a decrease in Mg⁺⁺ plasma level.

RESPONSE: Additional information has been included in the manuscript.

4. Data from VSMC mitochondria and from isolated liver mitochondria clearly demonstrated that Mg⁺⁺ improves mitochondria functional parameters. However, the manuscript lacks the direct evidence that Mg⁺⁺ enters into mitochondria. The reviewer wonders why the Authors did not quantify mitochondrial Mg⁺⁺ using specific fluorescent probes, as done by Yamanaka et al. 2016. Sci Rep 6: 30027.

RESPONSE: Mitochondrial magnesium has been now added in the revised version of the manuscript (new figures 4E and 8C).

5. Besides its several activities in mitochondria respiration, ATP production, oxidative stress..., Mg⁺⁺ is known to interact with lamins A, B and probably progerin (the three proteins bearing an Ig Fold interacting with Mg⁺⁺), with telomerase, all nuclear proteins known to contribute to progeria pathophysiological mechanisms. These aspects could be reported in the Discussion section.

RESPONSE: Additional sentences has been included in Discussion section.

Other questions/comments :

1. Despite some discrepancies between text and figures or between figure and legend (see below), the referee acknowledges the quality of figures as well as the pictures dissecting for the reader mechanisms, inhibitors...

2. Figure 1 :

• 1A : the black curve concerns WT VSMC. Perhaps this legend could be added above

the curve. The measurements begin at probably 5 days (?). Do 5 days for VSMC in culture is equivalent to passage 8 or 9 ?

RESPONSE: As we know, cell passaging, or splitting, is a technique that enables an individual to keep cells alive and growing under cultured conditions for extended periods of time. Cells should be passed when they are 90%-100% confluent, using a specific cells line split ratio (volume of flask surface area). Therefore, “passage” refers to number of cell divisions. In primary cell culture, the number of passage (number of cell divisions) is important due to the early loss of cell division capacity (compared to immortalized cells); and, therefore, the experiments need to be assessed in the same passage number. In the case of primary VSMCs, cells grow using split 1:3 and experiments are generally performed in passage 7-10. VSMCs lose replication capacity after passage 12-14, (depend on extraction procedure and culture method). Therefore, all the main experiments shown in this study were performed in the same passage (8 and 9). Figure 1 shows the cell division rate, which is different between cell type and conditions. In the case of Lmna^{G609/+} VSMCs needs more time to reach the confluence (1-2 additional days). Therefore, cells were frozen in different passages (with standard protocols) and they are thawed and grown when needed. In order to know the passage in figure 1, it is necessary to see number of cell (Y-coordinate). Values of 1, 2, 3 in Y-coordinate (as logarithmic) represent 10, 100 and 1000-fold cell number, which represent 3.3, 6.6 and 9.9 number of divisions. Experiment shown in figure 1B start at passage 8; Therefore, “30 days” for untreated LmnaG609G is equivalent to P12 (more and less). Figure 1 now includes “1” as cell number (Y-coordinate) at day 0 (X-coordinate). Additional information has been added in the Methods section and Figure legend.

• 1C to 1D : data from VSMC at 30 days in culture/passage 8 or 9 as written in the Material and Methods section ?

RESPONSE: Material and Methods section has been improved. All experiments were carried out in same passage (8 or 9).

• 1D (proliferation) and 1E (ATP) : a discrepancy with text page 5, lines 19 and 21 ; and page 16, line 12.

RESPONSE: Thanks for finding these errors.

• Perhaps to precise that asterisks close to the top of columns compare WT and untreated or treated cells

RESPONSE: Additional information has been included in Material Section, statistical analysis.

3. Figures 2, 3,4 : data from VSMC at 30 days in culture/passage 8 or 9 ?

RESPONSE: passages 8-9

4. Figure 4 :

- In manuscript, page 8, line 18, mitochondrial Ca⁺⁺ refers to figure 4C and not 4D

RESPONSE: Thanks for finding this error.

5. Figure 5 : the experimental protocol has to be better described in the Material & Methods section.

RESPONSE: Additional information has been included in the Material and Methods section and figure legend.

6. Figure 6 :

- 6B : what is the Mg⁺⁺ level in plasma from WT mice of the same genetic background and matched in sex and age ?

RESPONSE: Magnesium levels in plasma was included in table 1.

- 6D ; another discrepancy between text and figures : median survival time from 38.2 to 42.9 weeks in the text and more than 42 weeks and less than 48 weeks in the figure (n = 16 mice). These data have to be homogeneized in text and figure.

RESPONSE: A value of 43 weeks between 40 and 46 has been included in the X-coordinate.

- 6E : a comment (in Results or Discussion sections ?) regarding the marked decrease in body weight after 30 weeks ? Why mice escape from Mg⁺⁺ treatment ?

RESPONSE: Additional information has been included in Figure 6. Moreover, Magnesium treatment improved lifespan and body weigh by improvements in both mitochondrial function and mitochondrial ATP synthesis, and thus greater ATP availability, which is necessary for cellular energy supply and survival.

7. Figure 9 :

- 9B : another bug in 9B : legend quotes « Extramitochondrial NADH oxidation », whereas the Y axis of figure 9B indicates « Oxygen consumption ».

RESPONSE: Extramitochondrial oxidation of 1 mmol/L NADH was measured by oxygen consumption.

- 9C : text page 12 refers to Figure 9C missing in the corresponding figure.

RESPONSE: Thanks for finding this error.

In conclusion, this manuscript provided some convincing informations regarding the effects of Mg⁺⁺ in cultured cells and organs from G609G progeria mice model. However, the referee wonders if some results are not related to the very high quantity of Mg⁺⁺ intake by mice.

Moreover, the manuscript final sentence telling that Mg⁺⁺ dietary supplementation could benefit to HGPS children seems to the reviewer far from the clinical reality, at least because the too large quantity of Mg⁺⁺ was administered to mice in these experiments.

Finally, are missing, at least in the Discussion section, several data regarding Mg⁺⁺ plasma levels, dietary Mg⁺⁺ in (aged) man versus HGPS patients, in WT versus progeria mice, interaction of Mg⁺⁺ with lamins (a lack quite surprising knowing the main role of lamins in progeria) and their consequences.

For all these reasons related to its content, and because it has to be carefully improved in its form, the present manuscript is not suitable for publication in EMBO Molecular Medicine.

Referee #2 (Comments on Novelty/Model System for Author):

The model organism used in this study is possibly the best available mouse model of HGPS. In the heterozygous state (used in the submitted research), it also reproduces the heterozygous condition of human disease. I would prefer that the authors report the reason why they chose heterozygous rather than homozygous mice to perform their research.

RESPONSE: Heterozygous mice was used for two main reason: 1) HGPS children are heterozygous (as the reviewer has also indicated). And 2) vascular calcification is better observed in heterozygous mice than in homozygous mice because the formed mice live longer and thus there is more time for accumulate calcium. Moreover, homozygous mice consume dry food poorly, and their food is usually moistened with water so they can eat it properly. This makes it more difficult to control their magnesium intake. Although the same beneficial effects of magnesium can be observed in homozygous mice, magnesium should be delivered via daily injection rather than in food.

Referee #2 (Remarks for Author):

The manuscript by Dr. Villa-Belosta reports that Magnesium supplementation can improve the phenotype of progeroid vascular smooth muscle cells (VSMC) and extend lifespan in progeroid *LmnaG609G/+* mice. In particular, the author reports that magnesium treatment of VSMC improves ATP synthesis, reduces lactate accumulation and subsequent mitochondrial Calcium overload, reduces oxidative stress and increases cell viability.

Finally, the author shows that oral magnesium supplementation reduces in vivo VSMC calcification and increases lifespan of *LmnaG609G/+* mice.

The background and rationale of the study are strong and based on previous results obtained by the author's team and other research groups. Those results have been published in high impact journals.

Major concerns:

- All differences reported in the manuscript, though statistically significant, appear minimal in absolute values. Can the author state that they significantly impact on cellular senescence and organism ageing? Beta-Gal staining of cells and some pictures showing the animal phenotype could give a better idea of the effect of Magnesium supplementation.

RESPONSE: Beta-gal activity has been included in figure 1. A picture showing animal phenotype has been also included in figure 6.

- Several biochemical pathways are well described in the manuscript and suggest potential pathogenetic mechanisms. Could the author hypothesize (and test, if possible) how *Lmna G609G* mutation elicits the biochemical effects reported in the manuscript?

RESPONSE: A new figure 10 has been included showing the author`s hypothesis. As was indicated in discussion section, lamnaG609G/+ mutation sequesters the antioxidant Nrf2 pathway, which induce loss of antioxidant capacity and increments of ROS.

Referee #3 (Remarks for Author):

General comments on EMM-2020-12423 manuscript entitled: "Dietary magnesium supplementation improves lifespan in a mouse model of progeria "

Ricardo Villa-Bellosta investigates the impact of magnesium supplementation on vascular smooth muscle cells (VSMCs) and progeria (LmnaG609G/+) mice mitochondrial function, calcium deposits and lifespan. Using, a thorough analysis of several mitochondrial parameters including ATP synthesis, ROS levels, redox status, MMP and energy metabolism, the author provides evidence that magnesium treatment improves all these functions in both in vitro and in vivo studies.

Specific comments

Overall, the study is well designed, the details of the experiments and methodologies are clear and sufficient.

Results

-Paragraph: "Magnesium improves LmnaG609G/+ vascular smooth muscle cell (VSMC) viability"

Figure 1A shows the growth rate of LmnaG609G/+ VSMCs during a period of 30 days. A detailed description of this long-term culture is needed. For instance, how many passages were performed for each treatment condition during this 30-day period?

RESPONSE: As we know, cell passaging, or splitting, is a technique that enables an individual to keep cells alive and growing under cultured conditions for extended periods of time. Cells should be passed when they are 90%-100% confluent, using a specific cells line split ratio (volume of flask surface area). Therefore, "passage" refers to number of cell divisions. In primary cell culture, the number of passage (number of cell divisions) is important due to the early loss of cell division capacity (compared to immortalized cells); and, therefore, the experiments need to be assessed in the same passage number. In the case of primary VSMCs, cells grow using split 1:3 and experiments are generally performed in passage 7-10. VSMCs lose replication capacity after passage 12-14, (depend on extraction procedure and culture method). Therefore, all the main experiments shown in this study were performed in the same passage (8 and 9). Figure 1 shows the cell division rate, which is different between cell type and conditions. In the case of Lmna^{G609/+} VSMCs needs more time to reach the confluence (1-2 additional days). Therefore, cells were frozen in different passages (with standard protocols) and they are thawed and grown when needed. In order to know the passage in figure 1, it is necessary to see number of cell (Y-coordinate). Values of 1, 2, 3 in Y-coordinate (as logarithmic) represent 10, 100 and 1000-fold cell number, which represent 3.3, 6.6 and 9.9 number of divisions. Experiment shown in figure 1B start at passage 8; Therefore, "30 days" for untreated LmnaG609G is equivalent to P12 (more and less). Additional information has been added in the Methods section and Figure legend.

Hence, morphological images of the cell cultures at early and late time points should be included to visualize some potential changes. Moreover, the growth rate of mock-treated progeria cultures decreases more rapidly than control cells. Upon treatment with magnesium-supplement, the rate of growth increases in progeria VSCM cultures. How about normal VSCM cultures? Is the growth rate also ameliorated upon magnesium treatment? If yes this could suggest that magnesium supplement not only ameliorate progeria cells but possibly normal cells as well.

RESPONSE: Thank you very much for these interesting comments. Under our experimental conditions (addition of 1 mmol/L magnesium), no effect on growth rate was observed in normal VSMCs with magnesium-enriched media respect normal media. However, we do not rule out any effect with a higher magnesium concentration (addition of 2-3 mmol/L magnesium). Untreated WT mice were included in the manuscript to shown differences between WT and LMNAG609G/+ cells and mice. Finally, under or experimental conditions, no significative morphological changes were observed in VSMC during useful passes for experimentation. (usually P7-P10). For the primary VSMCs it is not recommended to use passages beyond passage 12 (P12). It is also not recommended to use below P5 because the cells need time to adapt to its ex vivo culture. Images of cell cultures are included to shown not differences between WT and LMNAG609G/+ VSMCs (Fig 1), or between treated and untreated LMNAG609G/+ VSMCs (Figure 5) under our experimental conditions.

Paragraph: "Magnesium prevents phosphate-induced LmnaG609G/+ VSMC calcification^[1,2]"

Figure 5: VSMC microscopy images are missing to appreciate the levels of calcification and the morphological changes occurring in progeria VSMC cells by comparison to wild-type cells.

RESPONSE: Microscopy images and pictures showing calcification, has been included in the figure 5. Not significant morphological changes can be observed correctly during calcification. Calcification consist in the deposition of calcium-phosphate crystals on the cells. When this occurs, it is not possible to see correctly cells because there are hydroxyapatite crystals in top of them. No morphological changes were observed during the early days.

Sentence in conclusion of the discussion section:" Moreover, several studies report an association between cardiovascular disease and the hardness of drinking water due to its differing magnesium content."

Please, rephrase this sentence. I don't understand the meaning.

RESPONSE: The sentence has been improved.

Beside the above comments, western blot analyses of the lamin A/C status are missing for both in vitro and in vivo studies. What is the impact of magnesium supplement on Lamin A/C levels?

RESPONSE: In our experimental conditions lamin A/C status is unaffected. However, magnesium could interact with nuclear proteins, including telomerase and lamins A, B and C, which could improve or reduce the reported beneficial effect of magnesium. These molecular mechanisms can be evaluated in future studies, including other metabolic pathways, signaling pathways and enzyme activities. Moreover, as has been widely commented in the discussion section, HGPS induce a repression in the antioxidant Nrf2 pathway, which are involved in expression of GR and glutathione synthesis. Our data shown no differences in GR activity and Glutathione Synthesis upon magnesium treatment. This data support not alteration at nuclear levels, including lamins ABC status, interactions and activities. However, all these facts are far from our main objective in this study: show the antioxidant properties of magnesium (including, improvement in ATP synthesis and reduction both in ROS and calcium overload) in HGPS mice. Finally, the antioxidant properties of magnesium have also been reported in different diseases/models without changes in the lamins A/C status. Additional information has been included in the Discussion section.

In closing, this is an interesting study. Hence, if magnesium supplementation is sufficient to restore mitochondrial function as indicated by the findings from this report, this opens a new perspective for treatment of HGPS patients and other conditions developing vascular disease.

7th Jul 2020

Dear Dr. Villa-Bellostá,

Thank you for the submission of your revised manuscript to EMBO Molecular Medicine. We have now received the enclosed report from the three referees who were asked to re-assess it. As you will see the referees are now supportive and I am pleased to inform you that we will be able to accept your manuscript pending the following amendments:

1. Please modify the discussion section according to referee 3's comment.
2. In the main manuscript file, please do the following:
 - Remove Figures from main manuscript file but leave the legends there.
 - Remove Web DOI's from references
 - Please remove the red color font
 - in legends, provide exact n= and exact p= values, not a range, along with the statistical test used. Some authors find that in order to keep the figures clear, providing an appendix supplemental table with all exact p-values is preferable. You are welcome to do this if you want to.
 - In Materials and Methods (and in the checklist), for animal work, confirm that all experiments were performed in accordance with relevant guidelines and regulations. The manuscript must include a statement in the Materials and Methods identifying the institutional and/or licensing committee approving the experiments. Gender, age and genetic background must be indicated, along with housing conditions.
3. Please check the figure callouts in the main article and make sure that all figures are called for. Currently Fig 5 A,B,C,D,E,G and Figure 6 F are not called out.
4. You have currently 10 figures, which is on the high side. We usually aim for 7-8 figures as main figures. Would you be able to identify a maximum of 5 figures to become Expanded view figure EV1-5? These figures are typeset like figures and called out in the text EV Fig. x. Legends for EV figures should be provided in the main text. EV Figures should be uploaded as figures, 1 / file, high resolution. More information can be found here:
<https://www.embopress.org/page/journal/17574684/authorguide#expandedview>
Please remember to update the figure callouts accordingly.
5. Please add scale bars in all microscope images.
6. The Paper Explained: EMBO Molecular Medicine articles are accompanied by a summary of the articles to emphasize the major findings in the paper and their medical implications for the non-specialist reader. Please provide a draft summary of your article highlighting
 - the medical issue you are addressing,
 - the results obtained and
 - their clinical impact.
 - This may be edited to ensure that readers understand the significance and context of the research. Please refer to any of our published articles for an example.
7. For More Information: There is space at the end of each article to list relevant web links for further consultation by our readers. Could you identify some relevant ones and provide such information as

well? Some examples are patient associations, relevant databases, OMIM/proteins/genes links, author's websites, etc...

8. Every published paper now includes a 'Synopsis' to further enhance discoverability. Synopses are displayed on the journal webpage and are freely accessible to all readers. They include a short stand first (maximum of 300 characters, including space) as well as 2-5 one sentence bullet points that summarize the paper. Please write the bullet points to summarize the key NEW findings. They should be designed to be complementary to the abstract - i.e. not repeat the same text. We encourage inclusion of key acronyms and quantitative information (maximum of 30 words / bullet point). Please use the passive voice. Please attach these in a separate file or send them by email, we will incorporate them accordingly.

Here are some examples:

<https://www.embopress.org/doi/10.15252/emmm.201911571>

<https://www.embopress.org/doi/10.15252/emmm.201910270>

<https://www.embopress.org/doi/10.15252/emmm.201911419>

Please also provide a striking image or visual abstract to illustrate your article. The image should be provided as a jpeg-format file, 550 px-wide x 400-600 px high.

9. As part of the EMBO Publications transparent editorial process initiative (see our Editorial at <http://embomolmed.embopress.org/content/2/9/329>), EMBO Molecular Medicine will publish online a Review Process File (RPF) to accompany accepted manuscripts.

a. In the event of acceptance, this file will be published in conjunction with your paper and will include the anonymous referee reports, your point-by-point response and all pertinent correspondence relating to the manuscript. Let us know if you do not agree with this.

10. Our data editor has made a couple of suggestions on your manuscript (see attached), please fix.

I look forward to seeing a revised version of your manuscript as soon as possible.

Sincerely,
Jingyi Hou

Jingyi Hou
Editor
EMBO Molecular Medicine

*** Instructions to submit your revised manuscript ***

*** PLEASE NOTE *** As part of the EMBO Publications transparent editorial process initiative (see our Editorial at <https://www.embopress.org/doi/pdf/10.1002/emmm.201000094>), EMBO Molecular

Medicine will publish online a Review Process File to accompany accepted manuscripts.

To submit your manuscript, please follow this link:

Link Not Available

- 1) a .docx formatted version of the manuscript text (including Figure legends and tables)
- 2) Separate figure files*
- 3) supplemental information as Expanded View and/or Appendix. Please carefully check the authors guidelines for formatting Expanded view and Appendix figures and tables at <https://www.embopress.org/page/journal/17574684/authorguide#expandedview>
- 4) a letter INCLUDING the reviewer's reports and your detailed responses to their comments (as Word file).
- 5) The paper explained: EMBO Molecular Medicine articles are accompanied by a summary of the articles to emphasize the major findings in the paper and their medical implications for the non-specialist reader. Please provide a draft summary of your article highlighting
 - the medical issue you are addressing,
 - the results obtained and
 - their clinical impact.This may be edited to ensure that readers understand the significance and context of the research. Please refer to any of our published articles for an example.
- 6) For more information: There is space at the end of each article to list relevant web links for further consultation by our readers. Could you identify some relevant ones and provide such information as well? Some examples are patient associations, relevant databases, OMIM/proteins/genes links, author's websites, etc...
- 7) Author contributions: the contribution of every author must be detailed in a separate section.
- 8) EMBO Molecular Medicine now requires a complete author checklist (<https://www.embopress.org/page/journal/17574684/authorguide>) to be submitted with all revised manuscripts. Please use the checklist as guideline for the sort of information we need WITHIN the manuscript. The checklist should only be filled with page numbers were the information can be found. This is particularly important for animal reporting, antibody dilutions (missing) and exact values and n that should be indicted instead of a range.
- 9) Every published paper now includes a 'Synopsis' to further enhance discoverability. Synopses are

displayed on the journal webpage and are freely accessible to all readers. They include a short stand first (maximum of 300 characters, including space) as well as 2-5 one sentence bullet points that summarise the paper. Please write the bullet points to summarise the key NEW findings. They should be designed to be complementary to the abstract - i.e. not repeat the same text. We encourage inclusion of key acronyms and quantitative information (maximum of 30 words / bullet point). Please use the passive voice. Please attach these in a separate file or send them by email, we will incorporate them accordingly.

You are also welcome to suggest a striking image or visual abstract to illustrate your article. If you do please provide a jpeg file 550 px-wide x 400-px high.

10) A Conflict of Interest statement should be provided in the main text

11) Please note that we now mandate that all corresponding authors list an ORCID digital identifier. This takes <90 seconds to complete. We encourage all authors to supply an ORCID identifier, which will be linked to their name for unambiguous name identification.

Currently, our records indicate that the ORCID for your account is 0000-0002-1680-552X.

Link Not Available

12) The system will prompt you to fill in your funding and payment information. This will allow Wiley to send you a quote for the article processing charge (APC) in case of acceptance. This quote takes into account any reduction or fee waivers that you may be eligible for. Authors do not need to pay any fees before their manuscript is accepted and transferred to our publisher.

Photos 400-800 DPI

*Additional important information regarding figures and illustrations can be found at <http://bit.ly/EMBOPressFigurePreparationGuideline>

The system will prompt you to fill in your funding and payment information. This will allow Wiley to send you a quote for the article processing charge (APC) in case of acceptance. This quote takes into account any reduction or fee waivers that you may be eligible for. Authors do not need to pay any fees before their manuscript is accepted and transferred to our publisher.

***** Reviewer's comments *****

Referee #1 (Remarks for Author):

The manuscript has been completed, corrected and improved as required by the referee. It can be published in EMBO Molecular Medicine.

Referee #2 (Comments on Novelty/Model System for Author):

The mouse model is adequate for the understanding of HGPS pathogenesis and treatment. In particular, the authors clearly explained the choice fo heterozygous mice in the reported study.

Referee #2 (Remarks for Author):

The authors have clearly presented all their results and the experimental strategy they used to explore the effect of Mg⁺⁺ supplementation in HGPS preclinical models. Data provided in the manuscript may pave the way to additional translational research and suggest therapeutic approaches for vascular calcification associated with premature and normal ageing.

Referee #3 (Remarks for Author):

I am satisfied with the corrections.

However, please modify the end of the discussion by clearly stipulating that further experiments are needed to test the effect of magnesium supplement in human HGPS context and validate the results obtained in mouse HGPS model..

The authors performed the requested editorial changes.

28th Jul 2020

Dear Dr. Villa-Bellostà,

Please find enclosed the final reports on your manuscript. We are pleased to inform you that your manuscript is accepted for publication and is now being sent to our publisher to be included in the next available issue of EMBO Molecular Medicine.

We would like to remind you that as part of the EMBO Publications transparent editorial process initiative, EMBO Molecular Medicine will publish a Review Process File online to accompany accepted manuscripts. If you do NOT want the file to be published or would like to exclude figures, please immediately inform the editorial office via e-mail.

Please read below for additional IMPORTANT information regarding your article, its publication and the production process.

Congratulations on your interesting work,

Jingyi Hou

Jingyi Hou
Editor
EMBO Molecular Medicine

Follow us on Twitter @EmboMolMed
Sign up for eTOCs at embopress.org/alertsfeeds

***** Reviewer's comments *****

*** ** IMPORTANT INFORMATION *** **

SPEED OF PUBLICATION

The journal aims for rapid publication of papers, using the advance online publication "Early View" to expedite the process: A properly copy-edited and formatted version will be published as "Early View" after the proofs have been corrected. Please help the Editors and publisher avoid delays by providing e-mail address(es), telephone and fax numbers at which author(s) can be contacted.

Should you be planning a Press Release on your article, please get in contact with embomolmed@wiley.com as early as possible, in order to coordinate publication and release dates.

LICENSE AND PAYMENT:

All articles published in EMBO Molecular Medicine are fully open access: immediately and freely available to read, download and share.

EMBO Molecular Medicine charges an article processing charge (APC) to cover the publication costs. You, as the corresponding author for this manuscript, should have already received a quote with the article processing fee separately. Please let us know in case this quote has not been received.

Once your article is at Wiley for editorial production you will receive an email from Wiley's Author Services system, which will ask you to log in and will present you with the publication license form for completion. Within the same system the publication fee can be paid by credit card, an invoice, pro forma invoice or purchase order can be requested.

Payment of the publication charge and the signed Open Access Agreement form must be received before the article can be published online.

PROOFS

You will receive the proofs by e-mail approximately 2 weeks after all relevant files have been sent to our Production Office. Please return them within 48 hours and if there should be any problems, please contact the production office at embopressproduction@wiley.com.

Please inform us if there is likely to be any difficulty in reaching you at the above address at that time. Failure to meet our deadlines may result in a delay of publication.

All further communications concerning your paper proofs should quote reference number EMM-2020-12423-V3 and be directed to the production office at embopressproduction@wiley.com.

Thank you,

Jingyi Hou
Editor
EMBO Molecular Medicine

Corresponding Author Name: Ricardo Villa-Bellosta

Journal Submitted to: EMBO mol med

Manuscript Number: EMM-2020-12423